# A network model of the barrel cortex combined with a differentiator detector reproduces features of the behavioral response to single-neuron stimulation

**Davide Bernardi**[1,2,3]*, **Guy Doron**[1], **Michael Brecht**[1], **Benjamin Lindner**[1,2]

**1** Bernstein Center for Computational Neuroscience Berlin, Berlin, Germany, **2** Institut für Physik, Humboldt-Universität zu Berlin, Berlin, Germany, **3** Center for Translational Neurophysiology of Speech and Communication, Fondazione Istituto Italiano di Tecnologia, Ferrara, Italy

* davide.bernardi@bccn-berlin.de

**Data Availability Statement:** Simulation code and data needed to reproduce all results can be downloaded from the following G-Node repository:

## Abstract

The stimulation of a single neuron in the rat somatosensory cortex can elicit a behavioral response. The probability of a behavioral response does not depend appreciably on the duration or intensity of a constant stimulation, whereas the response probability increases significantly upon injection of an irregular current. Biological mechanisms that can potentially suppress a constant input signal are present in the dynamics of both neurons and synapses and seem ideal candidates to explain these experimental findings. Here, we study a large network of integrate-and-fire neurons with several salient features of neuronal populations in the rat barrel cortex. The model includes cellular spike-frequency adaptation, experimentally constrained numbers and types of chemical synapses endowed with short-term plasticity, and gap junctions. Numerical simulations of this model indicate that cellular and synaptic adaptation mechanisms alone may not suffice to account for the experimental results if the local network activity is read out by an integrator. However, a circuit that approximates a differentiator can detect the single-cell stimulation with a reliability that barely depends on the length or intensity of the stimulus, but that increases when an irregular signal is used. This finding is in accordance with the experimental results obtained for the stimulation of a regularly-spiking excitatory cell.

## Author summary

It is widely assumed that only a large group of neurons can encode a stimulus or control behavior. This tenet of neuroscience has been challenged by experiments in which stimulating a single cortical neuron has had a measurable effect on an animal's behavior. Recently, theoretical studies have explored how a single-neuron stimulation could be detected in a large recurrent network. However, these studies missed essential biological mechanisms of cortical networks and are unable to explain more recent experiments in the barrel cortex. Here, to describe the stimulated brain area, we propose and study a

https://gin.g-node.org/davidebernardi/BerDorBreLin20-plos-CB-sub/wiki.

**Funding:** DB received funding from the German Research Foundation (DFG), GRK 1589/2 (https://www.eecs.tu-berlin.de/grk_15891). DB, GD, MB, and BL received funding from the German Federal Ministry of Education and Research (BMBF), Bernstein Center II (https://www.bccn-berlin.de) grant no. 01GQ1001A. The funders had no role in study design, data collection and analysis, decision to publish, or preparation of the manuscript.

**Competing interests:** The authors have declared that no competing interests exist.

network model endowed with many important biological features of the barrel cortex. Importantly, we also investigate different readout mechanisms, i.e. ways in which the stimulation effects can propagate to other brain areas. We show that a readout network which tracks rapid variations in the local network activity is in agreement with the experiments. Our model demonstrates a possible mechanism for how the stimulation of a single neuron translates into a signal at the population level, which is taken as a proxy of the animal's response. Our results illustrate the power of spiking neural networks to properly describe the effects of a single neuron's activity.

## Introduction

A classical method used in neuroscience to understand cortical circuits is to determine how single neurons respond to a controlled sensory stimulus. If, for instance, one of a rat's whiskers is moved by the experimenter, a change in firing of specific neurons can be measured in the barrel cortex, one of the most well studied parts of the primary sensory cortex [1]. The concept of "reverse physiology" turns this approach around by studying the inverse situation in which neurons in higher brain areas are stimulated and a behavioral or motor response can be elicited (see [2] for early references on such experiments). The two kinds of experiments in combination then allow the linking of sensation and perception, one of the notoriously difficult problems in neuroscience.

In the case that the stimulation affects only a single neuron, the outcome of the unconventional and technically challenging reverse physiology experiments are particularly striking: stimulating a single neuron in the motor cortex can evoke a whisker movement [3], and single-cell stimulation in the barrel cortex—but not in the thalamus—leads to a weak but statistically significant behavioral response [4–6]. This contradicts prevailing hypotheses that relevant signals can only be encoded in the activity of large neural populations.

Both the enormity of cortical networks—tens of thousands of neurons in the case of the somatosensory cortex [7]—and the apparent randomness of single-neuron spiking [8, 9] have classically been evoked as arguments for population coding. If single spikes are unpredictable and noisy how can a few externally induced spikes lead to changes in behavior?

On the theoretical side, cortical populations have been modeled as (locally) random networks of synaptically coupled excitatory and inhibitory cells [10–12] (many studies just take into account two distinct cell types). Even without the inclusion of explicit noise sources, these models can show asynchronous irregular activity [13–15] that is similar to that observed in the cortex of alert animals [16, 17]; this kind of network noise can also be described analytically by stochastic mean-field methods (see for instance [10, 13, 18–20]). Besides the autonomous activity of such networks, their linear and nonlinear response to global stimuli, i.e. applied to *all* neurons in the network, has been in focus [20–24]. However, injecting a current into a *single* neuron in such a generic network model can lead to sizable changes in a subpopulation's activity as well [25, 26]; this subpopulation can be regarded as a readout of the stimulus. If the readout population is somewhat oriented towards the direct postsynaptic partners of the stimulated cell, then the stimulus can be detected in the activity of this population [25]. If, more realistically, the readout is accomplished by a second recurrent network with feed-forward inhibition, as is most likely the case in the cortex, already a very small bias will lead to a detection performance comparable to that in the behavioral experiment [26].

Especially challenging for theoreticians are the results of the nanostimulation experiments in the barrel cortex of behaving rats [6] which demonstrated striking dependencies of the

behavioral response on the properties of the stimulating current. The response does not depend on the duration of the stimulus, it depends weakly on its intensity, and strongly on its irregularity: the response is greatly enhanced if the current varies irregularly within the stimulation window instead of being held constant. None of these findings can be explained by the generic setups previously investigated in the context of single-cell stimulation [25, 26]. The results in ref. [6] represent a challenging and complex set of constraints that are difficult to fulfill at the same time, especially for a model based on spiking network models, which are notoriously expensive to simulate and hard to treat analytically.

Here, we tackle this problem by studying a computational model that includes more biological details of the barrel cortex: a network with three distinct cell types (one excitatory neuron type and two distinct inhibitory interneuron classes), which are all modeled as integrate-and-fire neurons endowed with adaptation currents, short-term synaptic plasticity for chemical synapses, which can be facilitating or depressing according to the cell types, and electrical synapses (gap junctions). A further crucial difference to our previous studies is that we compare two ways of reading out the network's response to the stimulation of one randomly selected neuron: one based on the integration and one based on the differentiation of the network's activity. We also show that a basic excitatory-inhibitory circuit can be used to approximate the differentiator, and demonstrate that the response of this so-called *differentiator network readout* is in several key aspects similar to the behavioral response observed in the experiments in ref. [6].

## Results

The model consists of two parts: a recurrent network, in which a randomly selected excitatory regular spiking cell is stimulated to mimic the experiments, and a readout, which receives input from the recurrent network and can detect the stimulation.

### Recurrent network model

Fig 1 shows a scheme containing all essential features of our network model, briefly described in the figure caption. The recurrent network model represents the surroundings of the stimulated cell in a radius of about 200μm, in which connection probabilities can be considered as constant [27, 28]. Taking into account the estimated neuron density in the barrel cortex [7], a sphere with a radius of 200μm contains about $N = 2600$ neurons, which we therefore take as the network size. Although this network size corresponds to a fraction of one barrel, and an even smaller fraction of the entire barrel cortex, this part of the model will be referred to as "barrel cortex network" (BCN); we stress that the BCN only describes a generic subnetwork within the barrel cortex and is not tailored to one specific layer, in line with the single-cell stimulation experiments [4, 6], which did not target one layer in particular. The BCN consists of three neuronal populations: excitatory regular spiking cells (RS), inhibitory fast-spiking (FS) cells, and somatostatin-expressing low-threshold spiking (SOM-LTS) inhibitory neurons. These three cell types account for a large fraction of the neurons found in the barrel cortex (for instance, they account for about 99% of layer IV [29], but numbers in other layers are similar [30]). All neurons are modeled as one-compartment leaky integrate-and-fire (LIF) neurons. Besides leak conductance and spike generation, several other biological mechanisms are modeled, according to the cell type.

The largest population consists of 2000 excitatory RS neurons, which are sparsely connected to each other but densely connected to FS interneurons and SOM-LTS cells, as experimental studies report [29]. The membrane time constant of RS cells is lognormally distributed with mean $\tau_{m,e} = 20$ ms and a standard deviation of 20% of the mean [29, 31]. The 400 inhibitory FS

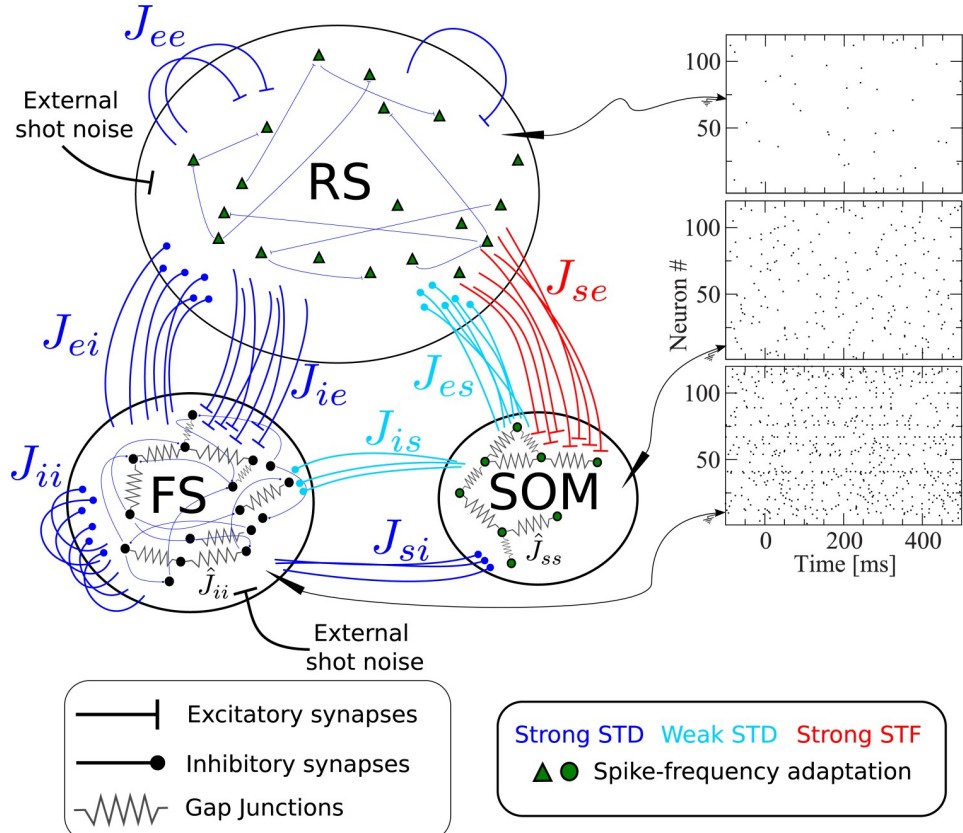

**Fig 1. Recurrent network model representing the surroundings of the stimulated cell.** The network is formed by $N_e$ = 2000 excitatory regular spiking (RS) neurons, $N_i$ = 400 inhibitory fast spiking (FS) neurons, and $N_s$ = 200 inhibitory somatostatin-positive low-threshold spiking (SOM-LTS) neurons. Recurrent connections between RS neurons are sparse (15%), all connections involving FS neurons as well as those between RS and SOM-LTS neurons are dense (40%-50%). FS and SOM-LTS neurons are electrically coupled (only neurons of the same type). Gap junctions are represented by an effective all-to-all spiking coupling (see main text). Connections in blue are strongly depressing, connections in light blue are weakly depressing, and connections in red are strongly facilitating. RS and SOM-LTS neurons are endowed with a spike-frequency adaptation current. Input from the thalamus and from neighboring cortical regions is represented by Poissonian shot noise. SOM-LTS neurons do not receive external shot noise. The three raster plots show the spontaneous activity of 120 (from top to bottom) RS, SOM, and FS neurons, respectively. The spontaneous activity of all three populations is asynchronous and irregular.

interneurons are characterized by faster membrane time constants (lognormal distribution with mean $\tau_{m,i}$ = 10 ms [29]) and are densely connected both to other FS neurons and to RS cells. The 200 SOM-LTS inhibitory neurons possess longer time scales (lognormal distribution with mean $\tau_{m,s}$ = 20 ms [29]) and a firing threshold which is 6 mV lower than in RS and FS neurons [29]. SOM-LTS neurons do not inhibit each other via chemical synapses, but form dense connections to and from RS neurons and sparser connections to and from FS interneurons [29, 32, 33].

Both FS and SOM-LTS neurons are densely coupled to cells of the same type via electrical synapses (gap junctions) [29, 32, 34], which are represented here by an effective global excitatory spiking coupling (the sub-threshold contribution of gap junctions has a much smaller effects on the network dynamics [35]).

In the barrel cortex, both RS and SOM-LTS neurons display spike-frequency adaptation, whereas FS neurons do not [29, 36]. Therefore, RS and SOM-LTS are endowed with a spike-

triggered hyperpolarizing current [37–39]. Consistent with experimental observations, the strength of the adaptation current is larger for RS neurons than for SOM-LTS neurons.

We drive RS and FS cells with Poissonian spike trains mimicking input from the thalamus and neighboring cortical areas. SOM-LTS cells are mostly subject to local input [29, 40] and therefore, in our model, they do not receive external input.

Experimental studies suggest that most synapses in the barrel cortex show depression, with the notable exception of connections from RS neurons to SOM-LTS cells, which have been found to be strongly facilitating [29, 41–43]. The short-term plasticity of chemical synapses is simulated here by means of a standard model [44, 45]. Parameters were chosen such that all synapses except those from and to SOM-LTS neurons are strongly depressing. These synapses are depicted in blue in Fig 1. Parameters for synapses branching from SOM-LTS neurons (represented in light blue in Fig 1) are set such that they display a weak depression. Finally, parameters of synapses connecting RS cells to SOM-LTS neurons are chosen to generate a strong facilitation. A further property that distinguishes these synapses is that the transmission failure rate is high (∼50%) at low presynaptic firing rate, but the reliability increases upon repeated activation of the synapse [29]. This property is modeled by a variable that mimics the activity-dependent failure rate.

More details on model equations and parameters are given in the Methods on p. 26.

## Readout models

We consider three readout schemes, as illustrated in Fig 2. The first detection scheme (Fig 2A) receives input from a subset of the BCN and reacts when the filtered activity of these neurons, which we will refer to as *readout activity*, reaches a lower barrier. This readout scheme will be called integrator readout (IR). The second readout scheme (Fig 2B) filters the readout activity in the same way as the IR, but it subtracts a time-shifted copy of the same activity. In other words, it considers the difference between the filtered activity at different time points, thus acting as a sort of differentiator. For this reason, it will be referred to as differentiator readout (DR). The third readout scheme (Fig 2C) is based on the summed activity of a second simple excitatory-inhibitory network of LIF neurons, which approximately implements the differentiation operation of the DR. Hence, we will call this third readout scheme differerentiator network readout (DNR).

**Integrator readout.** The first readout scheme, the integrator readout (IR), first filters the summed activity of the readout neurons, which are a subset of the BCN. This set of readout neurons can be divided into three subsets ($\mathcal{S}^{\mathrm{RS}}$, $\mathcal{S}^{\mathrm{FS}}$, and $\mathcal{S}^{\mathrm{SOM}}$), which are a random selection from the RS, FS, and SOM populations, respectively (see Fig 2A). Unless otherwise indicated, the size of the three readout sets is $N_{\mathrm{read}}^{\mathrm{RS}} = 1000$, $N_{\mathrm{read}}^{\mathrm{FS}} = 100$, and $N_{\mathrm{read}}^{\mathrm{SOM}} = 100$, respectively. The spike trains emitted by all neurons within the readout sets are linearly filtered by using the following dynamical equation:

$$\tau_{m,e} \frac{\mathrm{d}A_{\mathrm{ir}}(t)}{\mathrm{d}t} = -A_{\mathrm{ir}} + R_{m,read} \left[ \sum_{i \in \mathcal{S}^{\mathrm{RS}}} J_{read,i}^{e}(t) x_i(t) \right.$$

$$\left. - \sum_{i \in \mathcal{S}^{\mathrm{FS}}} J_{read,i}^{i}(t) x_i(t) - \sum_{i \in \mathcal{S}^{\mathrm{SOM}}} J_{read,i}^{s}(t) x_i(t) \right], \tag{1}$$

where $x_i$ is the spike train of the $i$th neuron within the readout set $\mathcal{S}^X$ (where $X$ = RS, FS, or SOM), the integration time constant is $\tau_{m,e} = 20$ ms, and $R_{m,read} = \tau_{m,e}/C_m$. Consistent with the idea that this readout is performed by other brain areas, and that synapses projecting to other

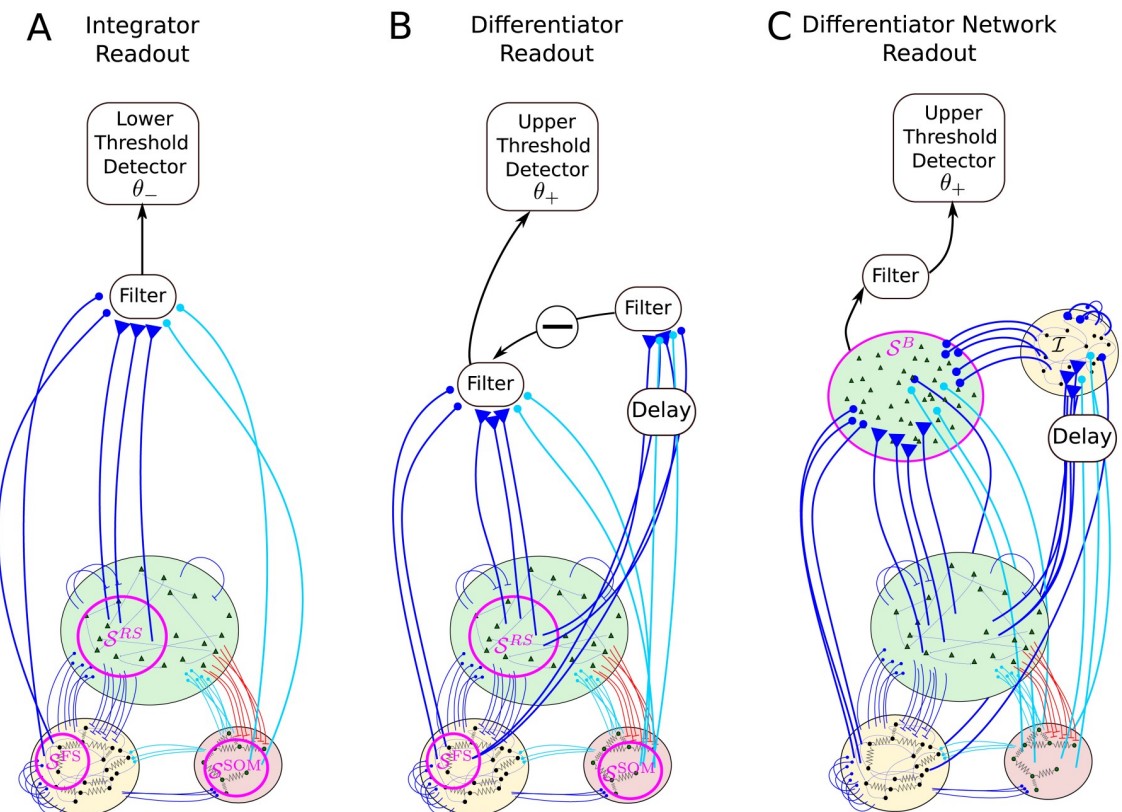

**Fig 2. Readout models considered in this paper.** One excitatory regular-spiking (RS) neuron from the barrel cortex network (BCN) is selected at random and stimulated. The BCN consists of three populations: RS neurons, inhibitory fast-spiking neurons (FS), and somatostatin-positive low-threshold spiking neurons (SOM-LTS), all modeled by leaky integrate-and-fire neurons. The BCN includes several biological details of the barrel cortex (see Fig 1 and Methods). Three readout schemes are considered. **A**: The integrator readout (IR) integrates the activity of a subset of the RS neurons within the BCN and reacts to deviations in the negative direction. **B**: The differentiator readout (DR) evaluates the difference between the IR activity at two time points separated by a delay. This filtered running difference at fixed lag is processed by the detector, which reacts when an upper threshold is reached. **C**: The differentiator network readout (DNR) approximates the operation of the DR with two populations of LIF neurons. The FS readout population ($\mathcal{I}$) provides delayed recurrent inhibition to itself and feed-forward inhibition to the RS readout population ($\mathcal{S}^B$), the activity of which is fed to the upper threshold detector. All connections depicted in blue and light blue are dynamic and show short-term depression (STD).

brain areas are likely to undergo short-term depression, the dynamic weights $J^e_{read,i}(t)$, $J^i_{read,i}(t)$, and $J^s_{read,i}(t)$ are depressing and obey the same equation as their counterparts within the BCN, i.e. Eqs (34) and (35); STD parameters are randomly distributed as those within the BCN, and depend on the cell type, so that $J^e_{read,i}(t)$ and $J^i_{read,i}(t)$ are more strongly depressing than $J^s_{read,i}(t)$ (see Methods). In other words, the readout activity $A_{ir}(t)$ can also be interpreted as the membrane potential of one "grandmother" neuron *without* fire-and-reset mechanism, which receives synaptic input from the readout sets only. This synaptic input is treated as any other synaptic input of the corresponding type in the network, with the only exception that readout weights from $\mathcal{S}^{SOM}$ to the readout are on average 20% stronger than those from the SOM to the RS population within the BCN.

To compute false positive and correct detection rates, a single lower decision boundary $\theta_-$ is used by the IR, as depicted in Fig 3A. A detection event is registered if $A_{ir}(t)$ falls at least once below the boundary $\theta_-$ within the detection time window $(0, T_w = 600\text{ms})$. In catch trials,

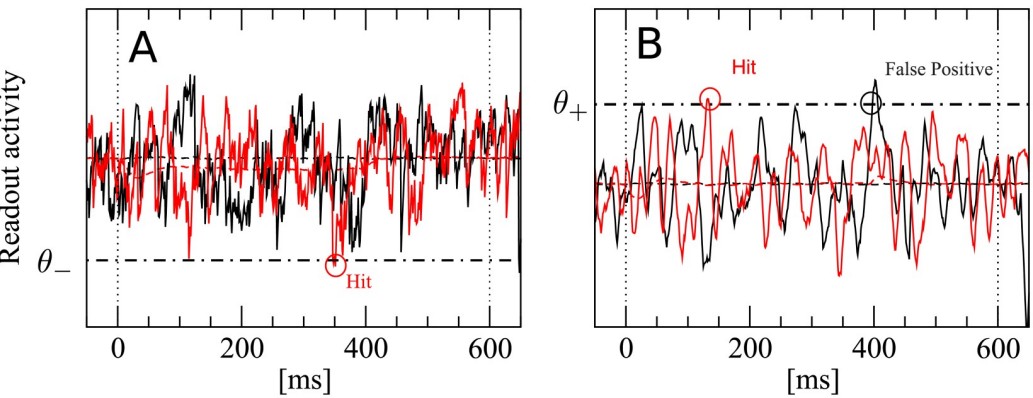

**Fig 3. A detection event is defined by a lower threshold crossing for the integrator readout (IR) and by an upper threshold crossing for the differentiator readout (DR) and the differentator network readout (DNR).** We show here an example for the IR (**A**) and the DR (**B**). **A**: The black trace represents one realization of the IR readout activity during one catch trial (i.e. in the absence of stimulus). The red trace represents one realization of the readout activity in the presence of a stimulus. **B**: Same for the DR. The readout activity of the IR and DR are defined in Eqs (1) and (4), respectively. The IR reacts whenever the readout activity falls at least once below the detection boundary $\theta_-$ (dashed-dotted line) within the detection time window (delimited by vertical dotted lines). The DR and the DNR react whenever the readout activity exceeds at least once the detection boundary $\theta_+$ (dashed-dotted line) within the detection time window (delimited by vertical dotted lines).

no stimulus is present. These catch trials are used to determine the *false positive rate*:

$$\mathcal{FP}_{\mathrm{ir}}(\theta_-) = \left\langle \max_{t \in (0, T_w)} \{H(\theta_- - A_{\mathrm{ir}}(t))\} \right\rangle_{\mathrm{catch}}, \tag{2}$$

where $H(x)$ is the Heaviside step function, and angular brackets indicate trial average. The reasons why a single lower detection boundary is used are explained below. The *hit* or *correct detection* rate is computed exactly in the same way, but in the presence of a stimulus

$$\mathcal{CD}_{\mathrm{ir}}(\theta_-) = \left\langle \max_{t \in (0, T_w)} \{H(\theta_- - A_{\mathrm{ir}}(t))\} \right\rangle_{\mathrm{stim.}}. \tag{3}$$

In Fig 3A, the red trace is a stimulus trial which falls below the chosen threshold. Hence, this trial is considered a correct detection. The black trace represents a catch trial. Because it does not cross the threshold within the detection window, no false positive event is registered in this trial. Note that the noise at the single-trial level is large.

**Differentiator readout.** The differentiator readout (DR) first reads in the input from the network in the same way as the IR and it takes the difference between $A_{\mathrm{ir}}$ evaluated at two times separated by a lag $\Delta T$. The result is then convolved with an exponential filter $\mathcal{F}_{\tau_f}(t) = \exp(-t/\tau_f)/\tau_f$ to reduce the noise

$$A_{\mathrm{dr}}(t) = (A_{\mathrm{ir}}(t) - A_{\mathrm{ir}}(t - \Delta T)) \star \mathcal{F}_{\tau_f}(t). \tag{4}$$

Trajectories computed from Eq (4) are used in combination with a simple threshold detector, as in the previous case. Note that the DR, however, uses an *upper* detection threshold $\theta_+$, as shown in Fig 3B. Black and red trials plotted in Fig 3B represent again a catch and stimulus trial, respectively. They both exceed the chosen threshold at least once within the detection window. Therefore, both trials contribute to false positive and correct detection rates, respectively. These rates are defined as in the previous case, except that here an upper threshold is

used:

$$\mathcal{FP}_{\mathrm{dr}}(\theta_+) = \left\langle \max_{t \in (0, T_w)} \{H(A_{\mathrm{dr}}(t) - \theta_+)\} \right\rangle_{\mathrm{catch}} \tag{5}$$

$$\mathcal{CD}_{\mathrm{dr}}(\theta_+) = \left\langle \max_{t \in (0, T_w)} \{H(A_{\mathrm{dr}}(t) - \theta_+)\} \right\rangle_{\mathrm{stim.}} . \tag{6}$$

There is nothing profound in the choice to endow the IR with a lower detection boundary and the DR with an upper detection boundary. The essential reason is that, in this way, we obtain a positive *effect size* (defined below), as it was observed in the experiments. In other words, this combination works for the readout schemes we consider here.

**Differentiator network readout.** The operation performed by the DR, i.e. the subtraction of a delayed copy of the readout activity, can be approximated by the simple network shown in Fig 2C. The differentiator network readout (DNR) consists of two populations: one readout population of 10000 RS neurons ($\mathcal{S}^B$) and one population of 2000 FS inhibitory neurons ($\mathcal{I}$). Each neuron within both populations receives the same number of feed-forward input connections from the three populations of the BCN as the size of the readout set of the respective cell type. More precisely, each neuron in the DNR receives input from $N_{\mathrm{read}}^{\mathrm{RS}} = 1000$ randomly chosen RS neurons, $N_{\mathrm{read}}^{\mathrm{FS}} = 100$ randomly chosen FS neurons, and $N_{\mathrm{read}}^{\mathrm{SOM}} = 100$ randomly chosen SOM neurons. Neurons in the readout population $\mathcal{S}^B$ evolve according to the same dynamical equation as RS neurons of the BCN, while neurons in $\mathcal{I}$ follow the same dynamical equations as the FS neurons of the BCN (see Methods).

One way to obtain a network that approximates a differentiator is that the mean input to the readout population $\mathcal{S}^B$ via the feed-forward inhibitory pathway from the BCN via $\mathcal{I}$ to $\mathcal{S}^B$ should cancel the direct feed-forward excitatory pathway input at a later time. To this end, the average weight of connections from $\mathcal{I}$ to $\mathcal{S}^B$, $J_{ei}^R$, should be chosen such that a static change in the input from the direct pathway $\Delta\mu_e$ is compensated by the static change in the input from the inhibitory pathway $\Delta\mu_{\mathcal{I}}$ (see Fig 4). The value of $J_{ei}^R$ that approximately satisfies the condition $\Delta\mu_e + \Delta\mu_{\mathcal{I}} \approx 0$ is computed from a linear-response calculation, (see p. 28). Importantly, the inhibitory pathway is given an additional transmission delay $\Delta T = 10$ ms, so that changes in the input from the BCN to the DNR are counterbalanced at a later time. As a matter of fact,

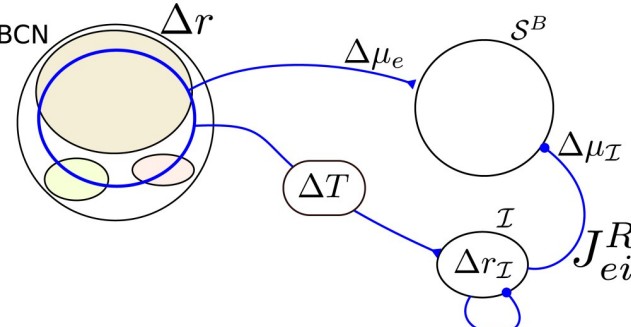

**Fig 4. Tuning of the differentiator readout network to implement the operation of the differentiator readout scheme.** A perturbation in the firing rate of the BCN ($\Delta r$) causes a perturbation in the mean input to the RS readout neurons, $\Delta\mu_e$, and a perturbation in the firing rate of the inhibitory readout population $\mathcal{I}$. This change in firing rate causes a shift in the input from $\mathcal{I}$ to $\mathcal{S}^B$ ($\Delta\mu_{\mathcal{I}}$). The strength of the connection from $\mathcal{I}$ to $\mathcal{S}^B$ ($J_{ei}^R$) is adjusted such that $\Delta\mu_e + \Delta\mu_{\mathcal{I}} \approx 0$. This cancellation reaches $\mathcal{S}^B$ with a time lag $\Delta T$ so that the readout network roughly implements the operation of the DR Eq (4).

choosing $J_{ei}^R$ such that the cancellation is not perfect grants an even better agreement with the experimental data, as shown below. More details on the DNR can be found in the Methods along with all parameter values.

The DNR activity is obtained by filtering the average firing rate of the readout neurons $\mathcal{S}^B$ with the same exponential filter used for the DR:

$$A_{\mathrm{dnr}}(t) = \frac{1}{N_B} \sum_{x_k \in \mathcal{S}^B} x_k(t) \star \mathcal{F}_{\tau_f}(t).$$

(7)

The DNR uses an upper detection boundary as the DR. Hence, false positive and hit rates are obtained in exactly the same way as done for the DR and depicted in Fig 3B):

$$\mathcal{FP}_{\mathrm{dnr}}(\theta_+) = \left\langle \max_{t \in (0, T_w)} \{ H(A_{\mathrm{dnr}}(t) - \theta_+) \} \right\rangle_{\mathrm{catch}},$$

(8)

$$\mathcal{CD}_{\mathrm{dnr}}(\theta_+) = \left\langle \max_{t \in (0, T_w)} \{ H(A_{\mathrm{dnr}}(t) - \theta_+) \} \right\rangle_{\mathrm{stim.}}.$$

(9)

## Effect size

The *effect size* is defined as the difference between the hit and the false positive rate [4]. It is a function of the detection threshold $\theta$

$$\mathcal{Y}_X(\theta) = \mathcal{CD}_X(\theta) - \mathcal{FP}_X(\theta),$$

(10)

where $X \in \{\text{ir}, \text{dr}, \text{dnr}\}$ indicates the detector type and $\theta$ can be either an upper or lower boundary. The false positive rate of 0.25 corresponds approximately to the average false positive rate measured experimentally. For this reason, this value was chosen to compare the simulation results to the experimental data. More precisely, the threshold $\bar{\theta}$ is chosen such that

$$\mathcal{FP}_X(\bar{\theta}) = 0.25,$$

(11)

which is then used to compute

$$\bar{\mathcal{Y}}_X = \mathcal{Y}_X(\bar{\theta}),$$

(12)

which is the final output of the detection procedure and will be compared to the experimental data.

## Single-cell stimulation

In every trial, the network is initialized with random initial conditions and simulated for $T_{\mathrm{idle}}$ = 1200 ms, to let the system forget the initial state. The spontaneous firing pattern of the network is asynchronous and irregular (Fig 1). The mean spontaneous firing rate of RS, FS, and SOM-LTS neurons is $r_{\mathrm{sp,e}} \approx 0.8$ Hz, $r_{\mathrm{sp,i}} \approx 10$ Hz, and $r_{\mathrm{sp,s}} \approx 3$ Hz, respectively. These properties of the spontaneous firing activity are consistent with experimental observations [46, 47].

A neuron is randomly selected as site of the nanostimulation, which is switched on at $t = 0$ and modeled as an additional current injected into the cell. The maximum stimulation current used here is $\Delta I_{\mathrm{max},e} = 5$ nA. This value is chosen to elicit a similar number of spikes as in the experiment and is in the range used experimentally [48]. After the stimulus is switched off, the network is further simulated until the time reaches $t = T_{end} = 1200$ ms.

Following [6, 48], we use step currents of different lengths and intensities to investigate how the response probability depend on the properties of the stimulus. Furthermore, random permutations of steps of different length and amplitude will be used to generate irregular spike trains. Two equally-sized sets of catch trials, i.e. trials in which no stimulus was present, were simulated to estimate the size of random fluctuations in the detection rates.

The shot noise mimicking external input and the initial conditions were drawn anew in every trial. The same realization of the network (randomized cellular parameters and the connectivity matrix including weights and delays) was used once for each stimulus type. The total number of trials per stimulus type was $N_{\text{trials}} = 10000$.

## Firing-rate response of the network

Before investigating to what extent the three readout procedures introduced above are capable to detect the single-cell stimulation, it is instructive to examine the trial-averaged firing rate response of the network to the stimulation of a single RS cell. In the following, the case of a constant step current with intensity at 25% of the maximum and a duration of 400 ms is considered.

When a single RS cell is stimulated (red triangle in Fig 5), its output spikes directly affect a relatively small set of RS neurons (blue shaded area in Fig 5) because RS-to-RS connections are sparse (15% connection probability). Furthermore, their average amplitude is smaller if compared to other connections and they are strongly depressing, so that—neglecting indirect multisynaptic paths—the *direct* excitatory effect on the overall firing rate of the RS population is small.

Connections from RS cells to the FS population are dense (40%), so that the spikes of the stimulated RS neuron directly reach a large fraction of the FS population (blue shaded area in Fig 5). However, FS cells also strongly inhibit one another and thus counteract the input from

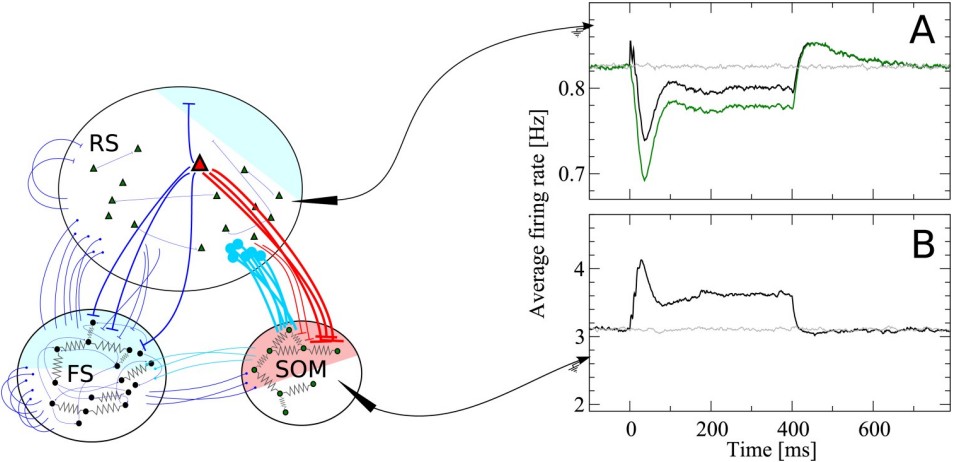

**Fig 5. Disynaptic inhibition mediated by somatostatin-positive low-threshold spiking (SOM-LTS) cells causes inhibitory response to the stimulation of a regular spiking (RS) cell.** When a RS cell is stimulated (the stimulus is switched here on at $t = 0$ and off at $t = 400$ ms), synapses from the stimulated cell (red triangle) to the SOM-LTS population strongly facilitate and cause a large increase in the firing rate of the SOM population, which then relaxes back to a plateau because of the spike-frequency adaptation (panel **B**, black line). The inhibitory input from the SOM to the RS population produces a response in the RS cells which is almost a mirror image (**A**, black line). The initial positive peak in the RS response is due to the spikes fired by the stimulated cell itself, as can be seen by excluding it from the population (**A**, green line). The curves in panels **A** and **B** are averages over 10000 trials and spikes are filtered with an exponential filter with decay constant $\tau_f = 15$ ms. Gray lines indicate catch trials (no stimulation). Colored shaded areas indicate the approximate fraction of the three populations which have a direct synaptic connection with the stimulated cell.

the stimulated cell. Consequently, the average change in the firing rate of the entire FS population is small.

Finally and most importantly, the output of the stimulated cell reaches a large fraction of the SOM-LTS population (50%, red shaded area in Fig 5) via strongly facilitating synapses. As a result, they induce an appreciable increase in the average firing rate of the SOM-LTS population, shown in Fig 5B. Importantly, SOM-LTS do not inhibit each other. However, the spike-frequency adaptation causes a strongly damped oscillation which, after an initial peak around $t$ = 30 ms and a dip around $t$ = 100 ms, relaxes to a plateau lying about 20% above the spontaneous firing rate level.

The increased inhibition from the SOM-LTS population ultimately causes the average firing rate of the RS population (Fig 5A, black line) to drop below the spontaneous level (Fig 5A, gray line). Note that the time course of the response of the RS population is closely related to that of the SOM-LTS population, except for the small peak shortly after the stimulus onset and for the overshoot after the stimulus is switched off. The small peak is due to the spikes fired by the stimulated cell itself. This can be seen by omitting the spikes fired by the stimulated cell (Fig 5A, green line) and observing that the peak disappears. The overshoot after $t$ = 400 ms is due to the slow relaxation of the adaptation variable to its baseline value.

Clearly, this description of how the single-neuron stimulation affects the firing rate of the network is greatly simplified, and it gives only a coarse picture of the actual network's response. However, these observations are consistent with *in vitro* experiments showing that the strong activation of a single pyramidal cell in the barrel cortex has mostly an inhibitory effect on nearby pyramidal cells, and that this effect is due to disynaptic inhibition mediated by SOM-LTS neurons [41, 42]. More recent *in vivo* experiments pairing the stimulation of a single cell with calcium imaging suggest that bursts induced in a pyramidal cell have a very weak excitatory effect on other pyramidal cells and on FS neurons, but have a significant effect on neighboring SOM-LTS cells [49], in line with the behavior of our model.

## Relation between effect size and statistics of readout activity

In the previous subsection, the effects of the single-cell stimulation on the *trial-averaged* response of the RS population have been examined. The readout, however, must decide on the presence of the stimulus based on the RS population activity *in each single trial*, which is a much more difficult task, as a comparison between the smooth lines of Fig 5A and 5B and the noisy curves in Fig 3 suggests.

The readout performance is quantified by the effect size, defined above. Before investigating how the effect size depends on the properties of the stimulus, we will examine how changes in the statistical properties of the readout activity $A_X(t)$ can influence the effect size.

The simplest statistics that can be considered are the time-dependent mean and standard deviation of the readout activity (the averaging ensemble consists of the different trials). Statistics of higher order (skewness and kurtosis) were measured and did not display appreciable deviations from the spontaneous state and will be therefore omitted for brevity. Because we are interested in deviations from the spontaneous state, it makes sense to consider mean and standard deviation of the readout activity as standardized deviations from the spontaneous value. More precisely, we will consider first the time-dependent deviation from the spontaneous value of the readout activity $A_X(t)$ (here $X$ = ir, dr, dnr, as defined above), normalized by the spontaneous standard deviation:

$$\hat{\mu}_X(t) = \frac{\langle A_X(t) \rangle - \mu_{X,\,\text{catch}}}{\sigma_{X,\,\text{catch}}}, \tag{13}$$

where $\mu_{X,\text{catch}}$ and $\sigma_{X,\text{catch}}$ are the average mean and standard deviation in the spontaneous state, respectively:

$$\mu_{X,\text{ catch}} = \langle A_X(t) \rangle_{\text{catch}}, \tag{14}$$

$$\sigma_{X,\text{ catch}} = \sqrt{\langle \Delta A_X^2(t) \rangle_{\text{catch}}}, \tag{15}$$

where $\Delta A_X^2(t) = (A_X(t) - \langle A_X(t) \rangle)^2$, and the time dependence in both last equations is self-averaging due to the stationary conditions. The time-dependent standard deviation of the readout activity is defined in a similar way (again a relative measure, given in multiples of the spontaneous standard deviation):

$$\hat{\sigma}_X(t) = \frac{\sqrt{\langle \Delta A_X^2(t) \rangle} - \sigma_{X,\text{ catch}}}{\sigma_{X,\text{ catch}}}. \tag{16}$$

Non-zero values of $\hat{\mu}_X(t)$ and of $\hat{\sigma}_X(t)$ at any time point within the detection time window can impact the effect size in different ways. Suppose, for instance, that the considered detector employs an upper boundary. Then, a positive deflection of $\hat{\mu}_X(t)$ locally increases the probability of reaching the threshold, whereas a negative deflection reduces it. If a lower detection boundary is used, the opposite holds. Regardless of the type of threshold, a local increase of $\hat{\sigma}_X(t)$ enhances the probability of reaching the threshold, whereas a local decrease in $\hat{\sigma}_X(t)$ reduces the probability of crossing the decision barrier. This line of reasoning is qualitative only and holds under the assumption that $A_X(t)$ is approximately normally distributed at all times.

To understand how multiple deviations from the spontaneous state within the decision time window jointly influence the effect size, it is useful to consider a simplified description of the decision model introduced in [25, 26, 50, 51]. In this simplified theory, hit and false positive rates are approximated as the result of $n = T_w/\tau_{\text{corr}}$ draws of a random variable, where $T_w$ is the detection time window and $\tau_{\text{corr}}$ is the autocorrelation time of the readout activity (in the example of Fig 6, $n = 4$). If these draws are treated as independent, the false positive rate reads

$$\mathcal{FP}(\theta) = 1 - p_0^n(\theta), \tag{17}$$

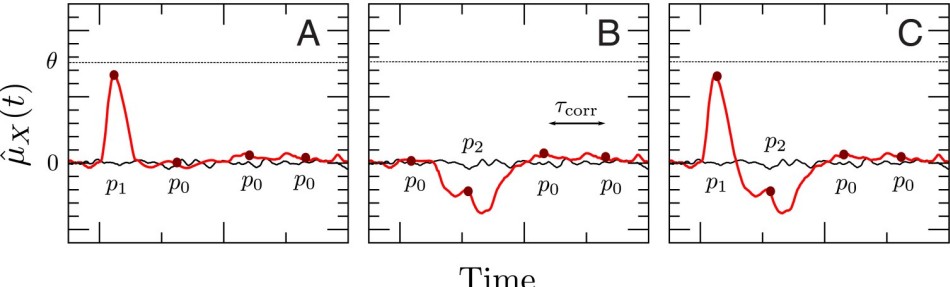

**Fig 6. Illustration of simplified detection model used to interpret the simulation results.** The continuous-time problem is approximated by a discretized process, obtained by "sampling" trajectories at times separated by the correlation time. In the spontaneous state, the probability that a time sample is *not* beyond the decision barrier $\theta$ is $p_0$. **A**: The peak decreases the local probability of not reaching $\theta$, i.e. $p_1 < p_0$ and the effect size is $\bar{\mathcal{Y}}_1$. **B**: The trough increases the local probability of not reaching $\theta$, i.e. $p_2 > p_0$; the effect size is here $\bar{\mathcal{Y}}_2$. **C**: in the case that both features are present and the changes in probability are small, the effect size is $\bar{\mathcal{Y}}_{12} \approx \bar{\mathcal{Y}}_1 + \bar{\mathcal{Y}}_2$, see text and Eq (22).

where $p_0(\theta)$ is the probability of *not* crossing the barrier $\theta$ at a given time point and in the absence of the stimulus [$p_0(\theta)$ does not depend on time and is therefore the same for each draw]. For concreteness, let us use an upper barrier at the value $\bar{\theta}$, which yields the false positive rate of 0.25. In this way, the dependence on $\theta$ can be dropped, but the following considerations do not depend on the particular position or type of the boundary. Suppose now that $\hat{\mu}_X(t)$ displays one peak at a certain position within the detection time window, as depicted in Fig 6A. Therefore, the probability that one trajectory of the readout activity triggers the detector is locally increased. Thus, in the vicinity of the peak, the probability of *not* triggering the detector will be $p_1 = p_0 + \Delta p_1 < p_0$. The correct detection rate for this situation is then

$$\bar{\mathcal{CD}}_1 = 1 - p_1 p_0^{n-1}. \tag{18}$$

Consequently, the effect size reads

$$\bar{\mathcal{Y}}_1 = 1 - p_1 p_0^{n-1} - (1 - p_0^n) = p_0^n \left( 1 - \frac{p_1}{p_0} \right). \tag{19}$$

Consider now the situation of a negative deflection in $\hat{\mu}_X(t)$ occurring at a different time (as in Fig 6B). Locally, the probability of not triggering the detector is $p_2 = p_0 + \Delta p_2 > p_0$. In this case, the effect size is

$$\bar{\mathcal{Y}}_2 = 1 - p_2 p_0^{n-1} - (1 - p_0^n) = p_0^n \left( 1 - \frac{p_2}{p_0} \right). \tag{20}$$

Suppose now that both features are present at sufficiently separated times within the same detection time window, as in Fig 6C. In this case, the effect size is

$$\bar{\mathcal{Y}}_{12} = p_0^n - p_1 p_2 p_0^{n-2} = p_0^n \left( 1 - \frac{p_1 p_2}{p_0^2} \right). \tag{21}$$

Substituting $p_1 = p_0 + \Delta p_1$ and $p_2 = p_0 + \Delta p_2$ into Eq (21) and supposing $\Delta p_1, \Delta p_2 \ll 1$ yields

$$\begin{aligned} \bar{\mathcal{Y}}_{12} &= p_0^n \left( 1 - \frac{p_0^2 + p_0 \Delta p_1 + p_0 \Delta p_2 + \Delta p_1 \Delta p_2 + p_0^2 - p_0^2}{p_0^2} \right) \\ &\approx p_0^n \left( 1 - \frac{p_0 + \Delta p_1}{p_0} + 1 - \frac{p_0 + \Delta p_2}{p_0} \right) \\ &= \bar{\mathcal{Y}}_1 + \bar{\mathcal{Y}}_2. \end{aligned} \tag{22}$$

This approximation generalizes to the case of more than two deviations from the spontaneous state [50], given that all deviations are small compared to $p_0$. For instance, when three features are present the effect size is

$$\bar{\mathcal{Y}}_{123} = p_0^n \left( 1 - \frac{p_1 p_2 p_3}{p_0^3} \right) \approx \bar{\mathcal{Y}}_1 + \bar{\mathcal{Y}}_2 + \bar{\mathcal{Y}}_3. \tag{23}$$

The main insight here is that weak deviations from the spontaneous state appearing in the same detection window can (approximately) add or cancel each other. This will be useful in the following to interpret how the behavior of $\hat{\mu}_X(t)$ and $\hat{\sigma}_X(t)$ influence the response of the detector.

## Detection of the stimulation of a single neuron

In this subsection, we will investigate how the properties of the stimulus influence the response probability of the detector for each of the three readout procedures.

**Effect of stimulus duration.** The effect of changing the stimulus duration will be considered first. To this end, stimuli of length 100, 200, and 400 ms are used (Fig 7A). The stimulus intensity is kept constant at 25% of the maximum current. In the experiment, when the stimulated cell was a RS neuron, the three stimuli evoked 6 ± 3, 11 ± 5, and 23 ± 10 spikes in the

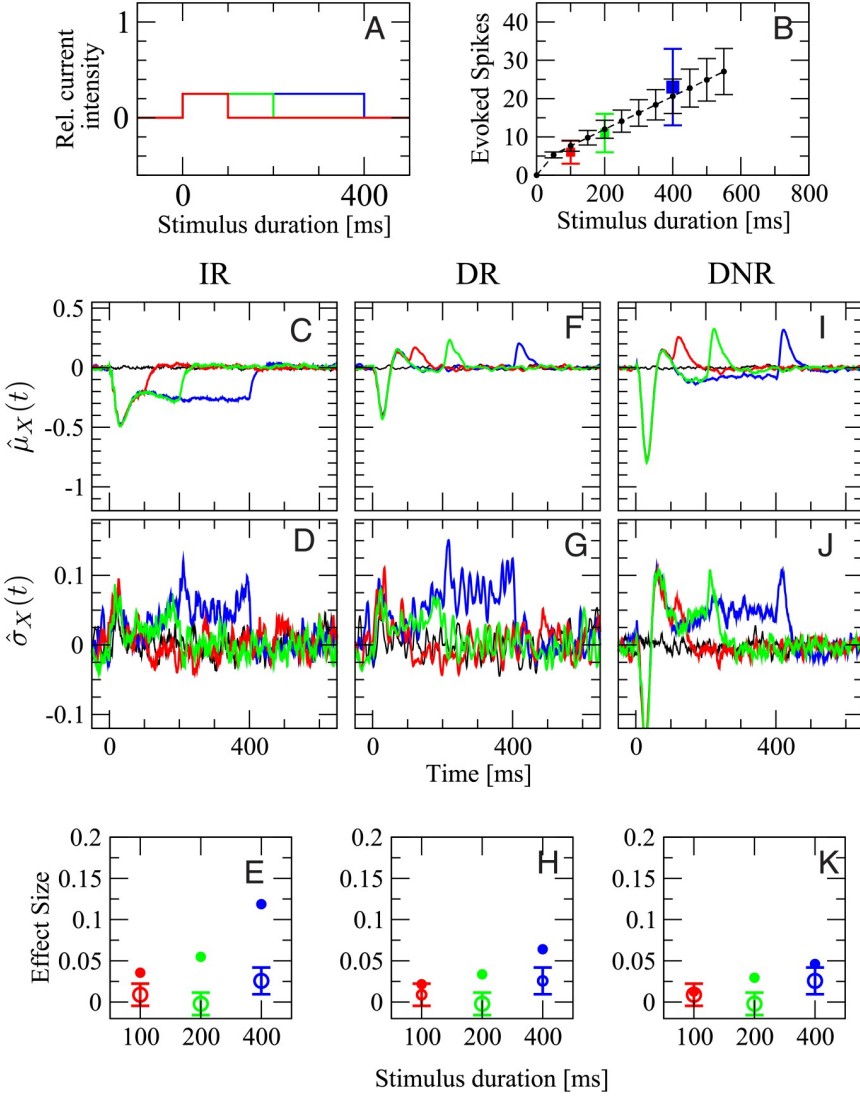

**Fig 7. Experimental data for stimuli of equal intensity and different duration are compatible with simulation results from DR and DNR, but not from IR. A**: stimulation currents used in this figure. The color coding (red: 100 ms, green: 200 ms, blue: 400 ms) applies to all panels. **B**: number of spikes evoked in the targeted neuron vs. stimulus duration in the model (black circles and dashed line) and in the experiments (squares colored as in **A**). First row (**C,F, I**): deviation from the spontaneous value of the time-dependent mean readout activity normalized by the spontaneous standard deviation, defined in Eq (13). Second row (**D,G,J**): deviation from the spontaneous value of the time-dependent standard deviation of the readout activity normalized by the spontaneous standard deviation, defined in Eq (16). Third row (**E,H,K**): average effect size for the three stimuli in **A**. Open circles with error bars: experimental data (average over 119 RS cells and 2407 total trials); filled circles of corresponding color: model simulations. First column (**C,D,E**): IR. Second column (**F,G,H**): DR. Third column (**I,J,K**): DNR. Black line: catch trial condition.

targeted neuron, respectively (Fig 7B, squares with different colors as in panel A). In the model, the number of evoked spikes was $7 \pm 1$, $12 \pm 2$, and $20 \pm 5$ spikes, respectively. The average number of evoked spikes generated by the model is within one standard deviation of the experimental data. However, the spread of the spike count distribution is smaller in the model, which is not surprising, considering the multiple possible noise sources that are not modeled, and that only some of the cellular parameters are randomly distributed in the model.

The remaining panels, Fig 7C–7K give an overview of $\hat{\mu}_X(t)$, $\hat{\sigma}_X(t)$, and the effect size measured by all three detectors with stimuli of different duration. In all plots, the color coding is as in Fig 7A and the black line represents catch trials. The various panels are organized as follows: the first, second, and third column represent results obtained from the IR, DR, and DNR, respectively; the first row shows the time-dependent trial average (expressed as standard score) $\hat{\mu}_X(t)$, the second row shows $\hat{\sigma}_X(t)$, and the third row displays the effect size defined as in Eq (12) (filled dots) superimposed with the experimental results (open circles with error bars).

The average response of the IR activity, $\hat{\mu}_{\mathrm{ir}}(t)$, is shown in Fig 7C for the three stimuli. It is a reduction that shows its deepest dip around $t = 40$ ms and then relaxes back owing to the spike-frequency adaptation of RS and SOM-LTS populations. Since a large part of the IR activity $A_{\mathrm{ir}}(t)$ is driven by the RS population and the further input from the SOM population has the same average postsynaptic effect (see Fig 5A and 5B and recall that the effect of the SOM population is inhibitory), it is not surprising that the time course of $\hat{\mu}_{\mathrm{ir}}(t)$ upon 400 ms stimulation (Fig 7C, blue line) closely resembles that of the RS population, i.e. the mirrored SOM response, shown in Fig 5. The disinhibitory input from the FS readout population does have an antagonist effect, but it is not sufficient to remove the average decrease in the IR activity. The different roles of the three readout populations will be discussed in more detail below (see p. 19).

In the case of the 100 ms stimulus (Fig 7C, red line), the mean deviation goes back to zero shortly after the stimulus is turned off, while $\hat{\mu}_{\mathrm{ir}}(t)$ for the other stimulus durations (green and blue line) settles around -0.3 for the remainder of the respective stimulation time window. The time-dependent standard deviation $\hat{\sigma}_{\mathrm{ir}}(t)$, shown in Fig 7D, displays a mild increase, especially in the later part of the stimuli. The time courses of $\hat{\mu}_{\mathrm{ir}}(t)$ and $\hat{\sigma}_{\mathrm{ir}}(t)$ suggest that the readout activity has more chances of reaching a lower detection threshold for longer lasting stimuli. Indeed, the effect size measured by the IR strongly depends on the stimulus duration, as can be seen in Fig 7E (filled circles), which is in contrast with the experimental data (open circles with error bars).

When the DR is used, the picture changes rather drastically. The time-dependent mean $\hat{\mu}_{\mathrm{dr}}(t)$ displays two peaks and one trough in response to all three signals (Fig 7F), even though the two peaks partly overlap in the case of the 100 ms stimulus (red line). Because the DR considers differences in the IR readout activity, each peak corresponds to an upswing of $\hat{\mu}_{\mathrm{ir}}(t)$ and the trough to the initial sharp drop. The most prominent feature in $\hat{\sigma}_{\mathrm{dr}}(t)$ is again a mild increase in the later part of the simulation time window (Fig 7G). The main difference in the response to the three stimuli is the position of the last peak. Hence, it stands to reason that the DR activity $A_{\mathrm{dr}}(t)$ has similar chances to reach the upper barrier regardless of the signal length. Indeed, the effect size measured by the DR displays only a weak dependence on the signal duration, due to the mild increase in $\hat{\sigma}_{\mathrm{dr}}(t)$ (Fig 7H).

The time course of $\hat{\mu}_{\mathrm{dnr}}(t)$ qualitatively resembles that of $\hat{\mu}_{\mathrm{dr}}(t)$ (see Fig 7I), thus suggesting that the DNR does approximately operate as a differentiator. The most evident difference between $\hat{\mu}_{\mathrm{dnr}}(t)$ and $\hat{\mu}_{\mathrm{dr}}(t)$ is the value of the plateau between the two peaks, which is slightly below the zero level. This negative response is due to the chosen tuning of the recurrent inhibition within the DNR and partly compensates the increase in the time-dependent standard deviation $\hat{\sigma}_{\mathrm{dnr}}(t)$, which behaves similarly to the mean, except that it is above the zero level in

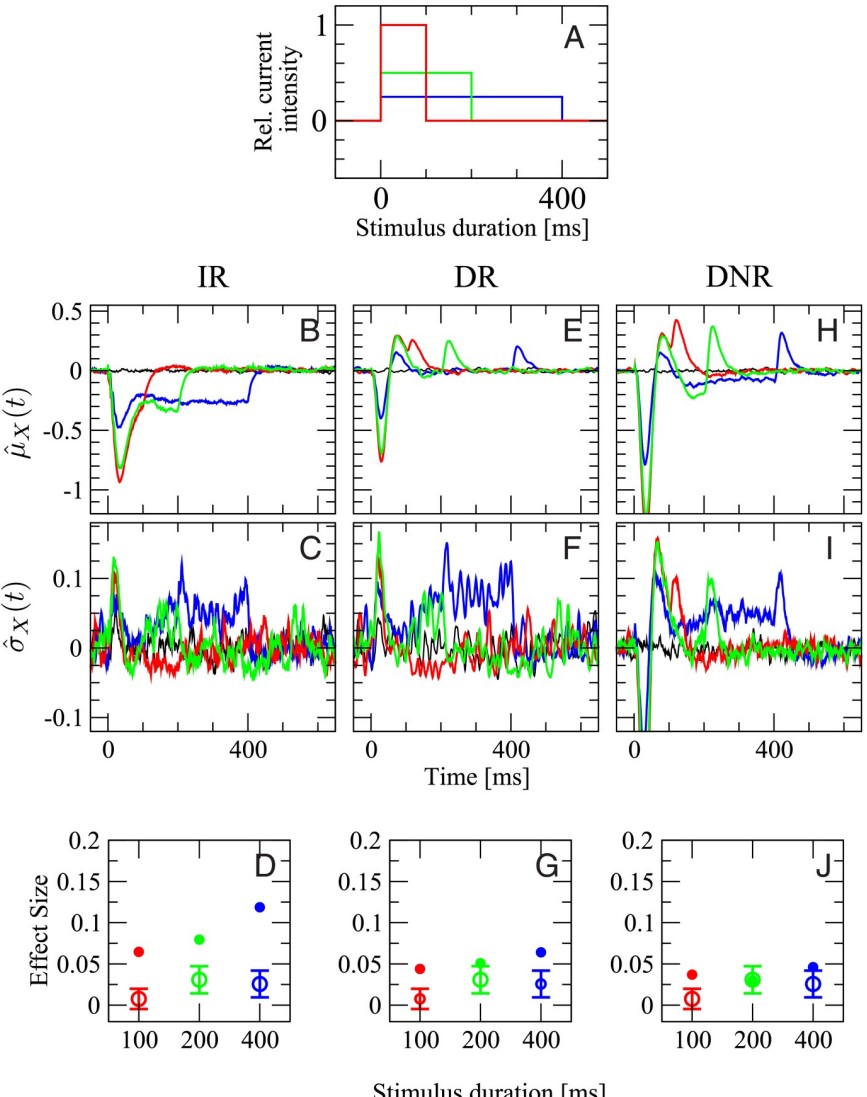

**Fig 8. Only results from DR and DNR are consistent with the experimental data for the stimulation with stimuli of intensity inversely proportional to the duration (signals are shown in A).** Panels **B**-**J** are organized as panels **C**-**K** of Fig 7 with the same color coding. Open circles with error bars in **D**, **G**, and **J** are the experimental average effect size measured from 55 RS cells (1469 total trials).

the central part of all stimuli (Fig 7J), and thus, it slightly increases the detection chance of longer stimuli. The combination of the two effects leads to an effect size that is barely dependent on the stimulus duration and that agrees rather well with the experimental results (Fig 7K).

To summarize the results of Fig 7, the effect size measured by the IR is larger but strongly depends on the duration of the stimulus. The DR and the DNR detect the stimulus with a reliability that is essentially independent of the signal duration and that is of the same magnitude as the experimental data.

**Effect of stimulus intensity.**    In the second experiment, we vary the firing rate of the stimulated cell by changing the current intensity while keeping the total area under the step stimulus, i.e. the injected charge, constant. As depicted in Fig 8A, the stimuli lasting 100 ms (red), 200 ms (green), and 400 ms have an intensity corresponding to 100%, 50%, and 25% of the

maximum current, respectively. In this way, the total number of elicited spikes is approximately the same for each stimulus. In the experiment, the three stimuli evoked a firing rate of $(109 \pm 52)$ Hz, $(54 \pm 23)$ Hz, and $(30 \pm 10)$ Hz, respectively. In the model, the average evoked rates are $(150 \pm 25)$ Hz, $(103 \pm 20)$ Hz $(50 \pm 12)$ Hz. Note that the maximum current in the model is chosen such that the number of elicited spikes roughly matches the data of the previous experiment (dependence on stimulus duration), which were based on a *different* set of cells. Still, the evoked firing rates in the model are in a similar range as the experimental values.

The remainder of Fig 8 reports all detection statistics arranged in the same way as before. The shape of $\hat{\mu}_{\mathrm{ir}}(t)$, the mean response of the IR to the three stimuli, is similar to the previous case, although the initial drop is much stronger here when the stronger stimuli are used (Fig 8B). A small peak in the time-dependent standard deviation $\hat{\sigma}_{\mathrm{ir}}(t)$ is observed right after the signal onset for the two stronger stimuli (Fig 8C). The pronounced initial response to the stronger signals partially compensates the shorter duration of the signal in terms of chances of reaching the detection barrier. As a result, the effect size measured by the IR for the two shorter, but stronger, stimuli is larger than in the previous case, and a clear dependence on the stimulus duration can be seen also in this case, as shown in Fig 8D.

The time-dependent mean of the DR activity, $\hat{\mu}_{\mathrm{dr}}(t)$, shown in Fig 8E, displays an initial drop followed by two peaks for each of the three stimuli. Here, however, both the first trough and the two subsequent peaks are more pronounced for signals of larger intensity. The behavior of $\hat{\sigma}_{\mathrm{dr}}(t)$ is similar to $\hat{\sigma}_{\mathrm{ir}}(t)$, and it marginally favors the detection of the longer signals (Fig 8F). As the most prominent features of $\hat{\mu}_{\mathrm{dr}}(t)$ have opposing effects and become stronger simultaneously upon growing stimulus intensity, the net effect on the effect size is barely noticeable (Fig 8G). Furthermore, the effect size is of magnitude comparable with the experimental observations.

Results obtained from the DNR are qualitatively similar to those from the DR. The principal differences are that both positive and negative deflections in $\hat{\mu}_{\mathrm{dnr}}(t)$ are more pronounced, and that the plateau between the two peaks is below the zero level (Fig 8H). The increase in $\hat{\sigma}_{\mathrm{dnr}}(t)$ is similar to that of $\hat{\sigma}_{\mathrm{dr}}(t)$ (Fig 8I). Finally, the effect size is essentially independent of the stimulus, as in the experiments (Fig 8J).

Hence, the effect measured by the DR and the DNR shows barely any dependence on the intensity of the stimulus, as it is observed in the experimental data. However, the effect size measured by the IR is larger and dependent on which of the three stimuli is used, which is not consistent with the data.

**Effect of stimulus regularity.**   In the third and last *in silico* experiment, random stimuli will be used to evoke irregular spike trains. These stimuli, in accordance with the experimental procedure [6, 48], are a random shuffling of six current steps of length 10, 20, 40, 80, 160, and 90 ms, and with current intensity 100%, 50%, 25%, 12.5%, 6.25%, and -50% of the maximum current, respectively. In other words, each sequence consists of a random permutation of five positive (depolarizing) current steps with intensity inversely proportional to the duration and of one hyperpolarizing step, which inhibits the cell from firing. Two example signals are shown in Fig 9A. Note that stimuli are varied in each trial and not frozen. The irregular stimuli are constructed such that their total duration is 400 ms. The response probability to these stimuli will be compared to that of regular steps of 400 ms at 25% of the maximum current, which was used in both previous cases (plotted in blue). In the experiments, irregular current injections generated spike trains with an average firing rate of $(24 \pm 11)$ Hz and average CV of $(1.1 \pm 0.3)$. In the model, the average rate is $(27 \pm 5)$ Hz and the average CV $(1.3 \pm 0.3)$.

In Fig 9 we compare the simulation results obtained when irregular stimuli are used to those obtained from the 400 ms regular current injection. The response to the latter is shown

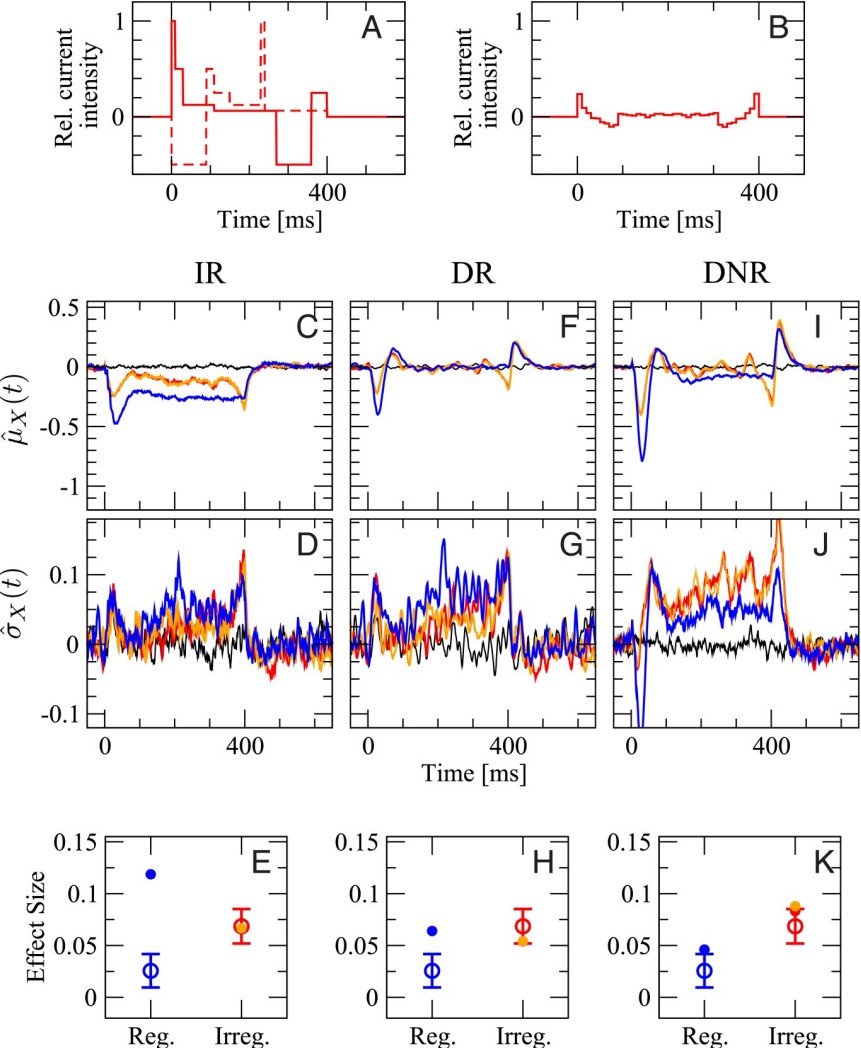

**Fig 9. In the comparison irregular vs. regular stimulation, simulation results from DNR agree well with the experimental data, whereas results from the IR are most inconsistent with the data.** Blue lines and symbols refer to the 400 ms regular stimulus, as in the previous cases; Red and orange lines and symbols refer to two different random samples of 10000 irregular stimuli (two specific realizations are shown in **A**, the average stimulus is shown in **B**; for details see text). Black line is catch trial condition (no stimulus). First row (**C,F,I**): deviation from the spontaneous value of the time-dependent mean readout activity normalized by the spontaneous standard deviation, defined in Eq (13). Second row (**D,G,J**): deviation from the spontaneous value of the time-dependent standard deviation of the readout activity normalized by the spontaneous standard deviation, defined in Eq (16). Third row (**E,H,K**): average effect size. Open circles with error bars: experimental data (average over 62 RS cells and 1780 total trials); filled circles of corresponding color: model simulations. First column (**C,D,E**): IR. Second column (**F,G,H**): DR. Third column (**I,J,K**): DNR.

once more in Fig 9 because it serves here as reference case but it will not be discussed in depth (see the discussion above). Results for irregular stimulation are based on two sets of irregular stimuli, constructed by choosing from all possible permutations with equal probability. Despite the large total number of trials used here ($N_{\text{trials}}$ = 10000), it is advisable to compare results for two independent sets of stimuli because the large number of possible permutations (6!=720) implies that the number of trials per signal is limited and that finite-size fluctuations due to the particular choice of signals may be non-negligible. Results for the two independent

sets of irregular stimuli are plotted in red and orange in Fig 9. We recall that $\hat{\mu}_X(t)$ and $\hat{\sigma}_X(t)$ are obtained by averaging over different realizations of the irregular stimuli and are not related to the two particular signals shown in Fig 9A.

The *average* signal is plotted in Fig 9B. By recalling that each irregular signal corresponds to one permutation of the step sequence, and recognizing that the reversed sequence is another valid and *equally probable* permutation, one can conclude that the average signal must possesses time-reversal symmetry. Indeed, it displays a peak just after the stimulus onset, followed by a mild trough and then by a plateau barely above the zero level. Just before the end of the stimulation time window, the symmetric trough followed by a peak can be seen. Accordingly, $\hat{\mu}_{\mathrm{ir}}$ shows two dips at the same time where the two peaks in the average signal are seen (Fig 9C). Note that although the first and second half of the average signal are perfectly symmetrical, the second dip in $\hat{\mu}_{\mathrm{ir}}(t)$ is more pronounced than the first.

Considering how different the average signal is from each particular realization of the irregular sequence, it may seem questionable to average over signals that provoke rather heterogeneous responses. However, the detector as well as the animals in the actual experiments do not know which realization of the irregular sequence is used in each trial. Therefore, this averaging ensemble, in which the stimulus is drawn in each trial, correctly represents the experimental situation and it makes sense to consider its time-dependent mean, as done above. The variability due to the particular realization of the input signal, which mostly averages out in the mean, reveals itself in an increased time-dependent standard deviation, $\hat{\sigma}_{\mathrm{ir}}(t)$ (Fig 9D), which is above the zero level in the entire stimulation time window and grows further towards the end of the stimulus. This increase of $\hat{\sigma}_{\mathrm{ir}}(t)$ above the zero level enhances the chances of reaching the detection threshold. As a result, the effect size upon irregular stimulation is as large as in the experiments, but not as large as that observed for the regular stimulus (Fig 9E filled dots), which is not consistent with the experimental observations, in which it is the other way around (Fig 9E, open circles with error bars).

The average DR activity in response to the irregular stimuli (Fig 9F, red and orange line) and to the 400 ms regular stimulus (blue line) are rather similar to each other. The main difference is that the initial trough and peak are somewhat smaller for irregular stimulation. Furthermore, a small dip is observed for irregular stimuli just before the last peak. Also the standard deviation $\hat{\sigma}_{\mathrm{dr}}(t)$ (Fig 9G) is similar in the two cases. Consequently, the average effect size upon irregular stimulation measured by the DR is similar for regular and irregular stimulation (Fig 9H, red and orange vs. blue full circles), which is in better agreement, but not completely consistent with the experimental observations.

The time-dependent mean DNR activity $\hat{\mu}_{\mathrm{dnr}}(t)$ (shown in Fig 9I) is qualitatively similar to $\hat{\mu}_{\mathrm{dr}}(t)$ although the peaks are more pronounced. Importantly, $\hat{\sigma}_{\mathrm{dnr}}(t)$ is generally larger than $\hat{\sigma}_{\mathrm{dr}}(t)$ in the case of irregular stimulation (Fig 9J) and displays a strong peak at the end. The larger $\hat{\sigma}_{\mathrm{dnr}}(t)$ sented (Fig 9K), which is as large as in the data.

In summary, Fig 9 shows that the DNR has the largest degree of consistency with the experimental observation that irregular stimuli are easier to detect than a regular current step, as opposed to the IR, which yields a smaller effect upon irregular stimulation than upon regular stimulation.

**Roles of cell types and delay difference on readout performance.**   After having shown that the DNR is most consistent with the experimental results, we will now explore how the readout performance depends on the size of the three readout populations, i.e. the effect of each neuron class on the detectability of the single-neuron stimulation. The simulation results are summarized in Fig 10. For brevity, we will focus on the IR (first row in Fig 10) and DNR (second row in Fig 10), and use only the 400 ms stimulus (represented by blue circles) and irregular stimulation (red squares).

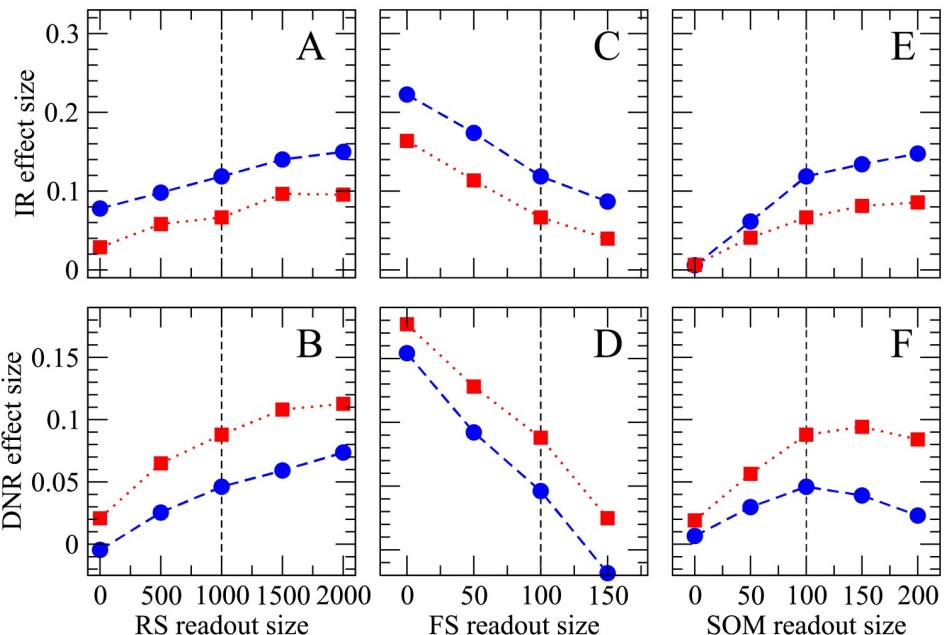

**Fig 10. Increasing the size of RS and SOM readout sets enhances the detectability of single-cell stimulation, whereas the size of the FS readout set has an opposing effect.** Effect size obtained upon 400 ms regular (blue circles) and irregular (red squares) stimulation as a function of the size of each readout population. The upper row (**A, C, E**) shows results for the IR readout, the lower row (**B, D, F**) displays the effect size measured from the DNR readout. The reference parameters used in the previous figures are indicated by the vertical dashed lines; in each panel the size of the readout population indicated on the x-axis is varied while all other parameters are kept fixed at their default value (as in Figs 7–9).

First, we can systematically vary the size of the RS readout population. Intuitively, the effect size should increase, the more RS neurons are fed into the readout. Simulation results demonstrate that this is indeed the case both for the IR readout (Fig 10A) and the DNR readout (Fig 10B): the effect size grows monotonically with the size of the RS readout population.

Next, we can study the effect of a systematic variation of the FS readout population's size. In Fig 10C and 10D), it can be seen how the effect size measured by the IR (DNR) decreases when the FS readout population is enlarged. The strong self-inhibition of FS cells tends to stabilize their firing rate. Hence, despite the strong SOM response within the BCN, the average firing rate of FS cells within the BCN is, on average, only slightly reduced during the single-cell stimulation. However, due to the strong output weights of FS cells, they have a considerable effect on the readout activity. Because they disinhibit the readout during the stimulation, the effect of FS cells on the readout is antagonist to that of RS cells.

The readout performance is improved by feeding more SOM neurons into the IR (Fig 10E). The readout performance of the DNR is generally enhanced by enlarging the number of SOM readout cells (Fig 10F). However, the effect size tends to saturate (for irregular stimulation) or even slightly decrease when the size the readout population approaches the entire SOM population (200 cells). One possible reason may be stronger cross-correlations between SOM cells.

As a final comment to Fig 10, we note that the DNR detects the irregular signal with a higher accuracy over the entire range of parameters, while the IR readout is almost always better at detecting the regular signal. In this respect, these findings are robust, and the DNR is a stronger candidate than the IR as a possible readout mechanism which is in line with the experimental results.

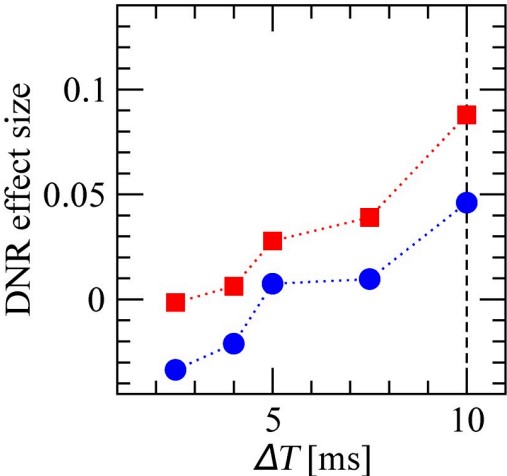

**Fig 11. Effect size measured by the DNR is robust for $\Delta T \gtrsim 5$ ms.** Effect size measured by the DNR as a function of the delay difference $\Delta T$ measured upon regular (blue circles) and irregular (red squares) stimulation.

One further crucial parameter for the operation of the DNR is the delay difference of the two input pathways ($\Delta T$). It is therefore important to check how robust our results are with respect to this parameter. As shown in Fig 11, the effect size measured upon irregular stimulation is larger than that observed for regular stimulation also for when $\Delta T$ is decreased below 10 ms, although it decreases in magnitude in both stimulus conditions. When $\Delta T$ is further decreased below 5 ms, the effect size reaches zero and becomes negative for the regular stimulus. Hence, the qualitative picture is robust for $\Delta T \gtrsim 5$ ms, as the weaker effect could be enhanced by leveraging, for instance, the dependencies seen in Fig 10.

We note here that it is difficult to try out all possible parameter combinations due to the high computational cost of simulating the network for multiple conditions and a large number of trials, which is needed to measure a possibly small effect size reliably. The simulation results shown in Figs 10 and 11 required about two weeks on a computing cluster with more than 200 cores.

**Robustness of readout mechanisms against slow input non-stationarity.** The above results show that the DNR is more consistent with the experimental results. However, the effect size measured by the IR is larger in magnitude in most cases, except for the case of irregular stimulation, in which the DNR is only slightly better. This observation raises the question of why the animal should opt for a detection strategy which tends to detect the signal less often, considering that experimental subjects were rewarded upon successful detection. One possible answer, which we will briefly explore in the following, is that the DNR might be more robust with respect to slow fluctuation of the background input. It is possible to get a feeling for these slow non-stationarities by looking at Fig 12A, which shows the spontaneous firing rate of some cells over long juxtacellular recording sessions. The strong changes in firing rate are mostly unrelated to any stimulation event. A readout that integrates spiking in a sliding time window might experience more difficulties in distinguishing the possibly small changes induced by the single-cell stimulation from the strong background variations, whereas a differentiator readout might still separate the timescales of stimulus and background noise.

Although a thorough investigation of this hypothesis goes beyond the scope of this study, in this last subsection we will explore this conjecture by adding a "static modulation" to the network background input. The basic idea is that a slow modulation of the network input will look nearly constant, if observed for the total duration of one of our trials ($\approx 2$ s). More

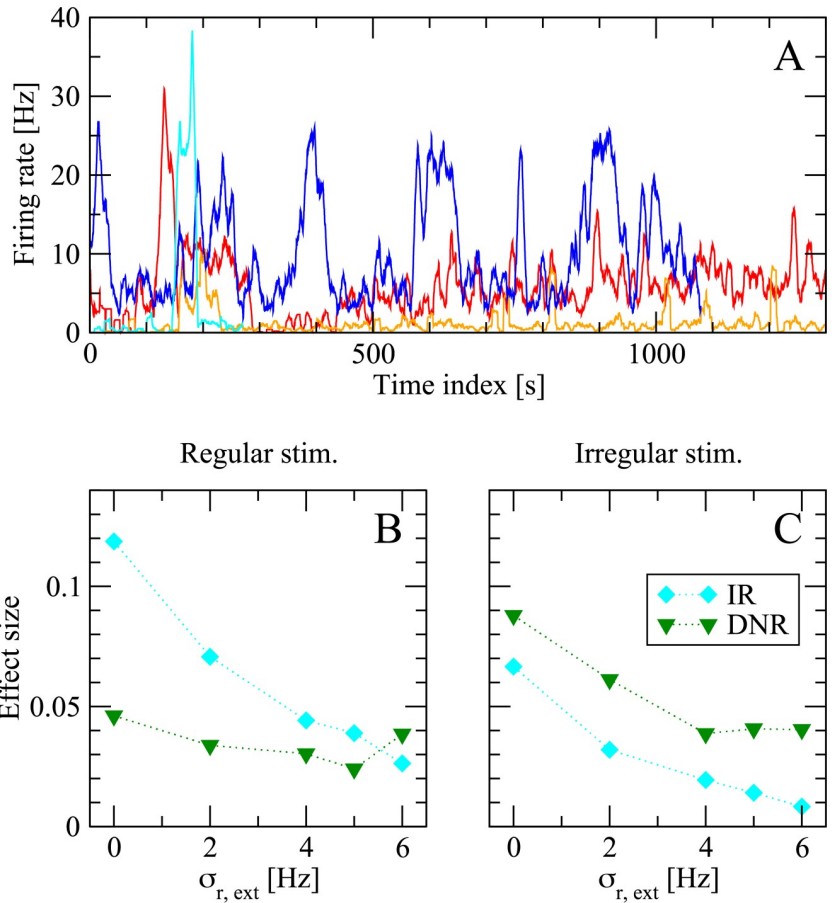

**Fig 12. The differentiator network readout (DNR) is more robust than the integrator readout (IR) against static readout noise, a proxy for slow modulations of the background network activity. A**: Firing rate of four different cells (filtered with a 1 sliding window) during juxtacellular recording sessions in the barrel cortex. The slow changes in firing rate are unrelated to stimulation events and give an example of slow non-stationarities present *in vivo* (note that the time window shown here is larger by orders of magnitude from the typical duration of a single trial in the detection experiments). **B**: Effect size measured in the model for the case of 400 ms stimulation on increasing $\sigma_{r,ext}$, the standard deviation of the global background input (input readout noise). Light blue diamonds represent results obtained from the integrator readout (IR), whereas green diamonds are results from the differentiator network readout (DNR). **C**: same as **B**, but for the case that irregular stimuli are used.

precisely, in each trial we randomly draw $r_{ext,th}$, one of the parameters that sets the global intensity of the external background noise (see Methods). Each sample is drawn from a lognormal distribution with constant mean (equal to 10 Hz, as in the simulations above) and standard deviation $\sigma_{r,ext}$ (where $\sigma_{r,ext} = 0$ corresponds to the situation considered so far, i.e. no background noise modulation). Fig 12B shows the effect size measured by the IR (light blue diamonds) and DNR (green triangles) upon increasing $\sigma_{r,ext}$ for the case of 400 ms regular stimulus. Both detectors are affected by the additional noise in the background input. However, the effect size measured by the IR drops faster and becomes eventually smaller than the effect measured by the DNR. For the case that an irregular stimulus is applied, Fig 12C shows that both detectors suffer a performance loss when $\sigma_{r,ext}$ is increased. However, the performance loss is stronger for the IR than for the DNR, which measures a larger effect size over the entire range of $\sigma_{r,ext}$ considered here.

## Discussion

The development of juxtacellular stimulation has brought remarkable experimental opportunities, ranging from reliably evoking prescribed spike trains [52–54] to probing the role of a single neuron in the perception of sensory inputs and motor responses [2–6]. Although these experiments question the well-established belief that large numbers of neurons are needed to encode a stimulus or elicit a behavioral response, they have received little attention in the theoretical and modeling literature. Recently, first attempts were made to model the behavioral responses to the stimulation of a single cortical neuron in rodents [25, 26, 55]. However, how the behavioral response probability is influenced by the properties of the injected current [6, 48] is still an open theoretical challenge. A recent computational study has demonstrated how irregular stimuli can be more effective in triggering a network burst than a regular stimulus [55]. However, the proposed mechanism does not provide an explanation for the weak sensitivity to the stimulus duration or intensity. In the present study, we constructed a model on different premises, in which the network keeps firing asynchronously and does not burst. Our model can successfully reproduce that the probability of the behavioral response is not substantially influenced by the duration or by the intensity of a constant stimulus, but it is strongly dependent on whether irregular or regular stimuli are used.

The spiking network model (BCN) we constructed in this study incorporates several features of the barrel cortex, and its parameters were consistent with the experimental literature. The BCN was chosen to represent only the immediate surroundings of the stimulated cell and its size is about one sixth of one standard cortical column [7]. Being too small, our network does not have a layered structure. Furthermore, it does not represent one specific layer within one "cortical column" because there was no clear layer specificity from the experimental side, and, presumably, cells from different layers were stimulated. Also the readout part of our model represents a generic downstream area and may or may not be located within a specific layer, or within the barrel cortex itself. One hypothesis could be that the stimulated cell resides in deeper layers and that the readout network is located within the supragranular layers. At the present state, however, there is no specific element in support of this conjecture before others, including the possibility that the readout computation is not localized in a single population, but rather distributed across several downstream areas.

It is still unclear whether specific anatomical properties of the barrel cortex make it particularly sensitive to the single-neuron stimulation. In this regard, although we did use the available experimental literature on the rat barrel cortex as reference to choose the range in which all cellular and synaptic parameters are heterogeneously distributed, some of these characteristics are similar in other cortical areas. Hence, the general traits of the BCN could describe also other cortical systems. If this is the case, other cortical areas might be sensitive to the single-cell stimulation.

The increased response probability for an irregular stimulation and the weak sensitivity to a step-current bring to mind the much-debated observation that irregular stimuli increase the response consistency of a single cortical cell *in vitro* [56], which was recently observed also *in vivo* [52]. These studies, however, considered the reliability in the spike sequence generated by a single cortical cell in response to repeated presentation of a frozen stimulus, and not the response to randomized stimuli at the behavioral level. Hence, it was still a completely open question what kind of mechanisms could be involved in the response of the local network and in the subsequent processing stages to regular and irregular stimuli. Among the biological processes included in the model, short-term depression and spike frequency adaptation could be expected to oppose slow changes in the input. However, our results indicate that these mechanisms may not suffice to explain the data if an integrator readout (IR) is employed. If a

differentiator readout (DR or DNR) is used instead, simulation results are in agreement with the data. In this respect, our results fit into the picture emerging from the classic experimental and modeling studies showing that the barrel system as a whole responds to whisker movements more as a differentiator rather than as an integrator [57–66]. These studies showed with an elegant combination of experiments and computational modeling how the on- and off-responses to whisker deflections, which are already present in the thalamic response, are sharpened within the barrel cortex, and elucidated the key role of inhibition, and, in particular, of cross-whisker inhibition. Refs. [57–66] already established the key role of inhibitory neurons in making the barrel system respond more like a differentiator than an integrator, and they did not postulate the existence of a further readout circuit, which performs the differentiation. Our model suggest that at least one further processing stage is needed to explain the experiments from ref. [6], which, in these earlier work, was not needed to explain the response properties of the barrel cortex. It must be noted, however, that we consider here a completely different situation, i.e. that of an extremely weak (at the network level) input that is delivered *directly into the cortex*. This stimulus, therefore, overrides the preprocessing performed by the afferent sensory pathways, which contribute significantly to the differentiation operation performed by the whisker sensory system. It was, hence, a still open question whether a similar kind of sensory processing that applies to a *volley of correlated inputs* such as those received from thalamocortical projections upon whisker deflection [57, 65, 67, 68] would apply to the experimental situation of refs. [4, 6]. Our model suggests that this might be the case, provided that a second downstream area acts as a differentiator. The involvement of further processing stages is also reasonable if one takes into account that the barrel cortex is part of a primary sensory area, and that it is, therefore, implausible that it can encode a decision or elicit a motor response directly. It is, indeed, likely that at least one more processing stage is involved in the difficult task of detecting and reporting the single-neuron stimulation.

One way of identifying candidate areas as a readout network would be an experiment in which the juxtacellular stimulation is performed in parallel with multi-electrode array extracellular recordings of the pooled activity of the local network. In this way, one could directly access $\hat{\mu}_X(t)$ and $\hat{\sigma}_X(t)$ (the deviation from the spontaneous state of the pooled local activity's mean and standard deviation, and thus extend the experimental data that can be compared to our model results beyond the change in the probability of a behavioral response, i.e. the effect size.

Beside being closer to the normal operation of the whisker system, a further argument for why the readout should differentiate the local network activity is provided by our results shown in Fig 12 and discussed on p. 21: there, we introduced a proxy for slow modulations of the input to the network, i.e. a static input noise. Our results showed that, in this simplified situation, the readout performance of the integrator readout is more severely degraded than that of the differentiator. Although it remains to be seen whether this finding holds also for a true dynamical input noise, differentiating would be a strategy that the detector uses to deal with strong and slow variations in the network's firing state.

The circuit configuration we chose for the DNR hinges on a difference in transmission delays, which in the standard case was $\Delta T = 10$ ms. This choice was made because it roughly matches the timescales of the changes induced in the readout population by the current jumps and thus ensures a good signal-to-noise ratio. Is this value physiologically plausible? The intersomatic distance required to achieve such a time difference in the barrel cortex would be approximately 2mm [69], which is a large but not unphysiological value [70]. As a matter of fact, Fig 11 shows that this value could be halved without compromising the qualitative picture. Moreover, although in the model the additional delay was entirely assigned to the connections from the BCN to the inhibitory readout population $\mathcal{I}$ (Fig 2C), it would be possible to

distribute the total latency among these connections and those from $\mathcal{I}$ to the readout population $\mathcal{S}^B$. In this way, the disynaptic inhibitory pathway would need, for instance, to travel back and forth over an even shorter distance.

Except for some minor technical differences, interpreting the IR activity as the voltage of a single "grandmother" neuron renders the detection problem somewhat reminiscent of the "Tempotron" readout [71], which can learn to respond to specific spiking patterns. Theoretical studies have shown that one single spike can be enough to perturb the firing pattern of a spiking network [72, 73], and it has been argued that the same may hold for the actual cortex as well [9]. Since the nanostimulation elicits multiple spikes in the targeted neuron, it is likely that some learning algorithm sensitive to precise spiking patterns, akin to the Tempotron, could easily detect the single-cell stimulation and achieve a better performance than our readout, which simply considers the pooled activity from a fraction of the local network activity. In fact, it has been shown that a simple classifier can discriminate highly correlated inputs to a spiking network which exhibits chaotic spontaneous firing activity [74]. Although some explicit training of the readout weights could indeed drastically increase the effect size and the experimental subjects did undergo a training phase before the single-neuron stimulation sessions [4, 6], we chose not to explicitly train the readout to detect specific cells, because the training in experiments was performed by employing microstimulation pulses, which intricately affect a large area rather than a specific cell directly [75–78]. Our readouts, that rely on the pooled activity of the local network, discard a lot of information that can be stored in precise spiking patterns, but they are also more robust and, most importantly, insensitive to the choice of the specific neuron.

In the construction of our model, we implicitly assumed that the training had already occurred and that it resulted in the formation of the differentiator circuit. It is possible that training the readout to detect the microstimulation with a suitable learning rule would produce a detector that is also more efficient in detecting the single-cell stimulation than the simple differentiator considered here. To this end, one could employ, for instance, a Tempotron-like rule, as mentioned above, or a Hebbian paradigm that pairs the microstimulation with the response of the readout population connected over several possible paths to the BCN. Beside the learning rule, however, an important problem would be how to construct a proper model for the complex, and only partially understood, cascade of events triggered by cortical microstimulation.

In our model, the main effect of inducing sustained firing in a single excitatory cell was to recruit SOM-LTS cells, which, in turn, inhibited the surrounding excitatory neurons. These results are consistent with the disynaptic inhibition observed *in vitro* [41, 42] and with recordings under anesthesia showing that bursts in pyramidal cells mostly activate surrounding SOM cells, hardly affecting other pyramidal cells or neighboring fast-spiking neurons [49].

Many different classes of inhibitory interneurons have been identified in the neocortex [79]. In this study we decided to include only two types of interneurons to try to limit the already high degree of complexity of the model. Modeling PV neurons is necessary as they are the most common interneuron type and form the backbone of the inhibitory system. SOM-LTS cells were included both because they are the second most common type of interneuron in the barrel cortex [30] and because other experimental studies hinted at their possible functional role when a single pyramidal cell is firing at high rates [41, 42, 49]. Another important class of cortical interneurons is formed by vasoactive intestinal peptide (VIP) neurons. These neurons do not directly provide inhibitory input to pyramidal cells and receive comparatively weak input from pyramidal cells. However, they have been found to make connections to and receive connections from SOM-LTS cells [33]. A recent computational study shows

that the mutual inhibition between VIP and SOM-LTS cells can modulate the response of pyramidal cells to external input [80]. On this basis, it may be speculated that VIP neurons also amplify the response to the single-cell stimulation through disinhibition. In other words, SOM-LTS cell activation would inhibit VIP neurons which, in turn, would disinhibit SOM-LTS and amplify the effects of single-cell stimulation. Because VIP neurons are believed to receive top-down input, this conjecture would explain how the attention level of the experimental subjects positively influences the ability to detect the single-cell stimulation [4, 6].

The network model considered here represents the surroundings of the stimulated cells, which justifies the choice of a random unstructured connection topology within each neuronal population. Expanding the model beyond the local scale requires a structured or distance-dependent connectivity profile. Spatial connection profiles and non-random topologies have strong repercussions on the cross-correlations between the spike trains in a network [81–84]. Cross-correlations, in turn, largely contribute to the fluctuations in the pooled activity of a large readout population, as was considered here [25, 50, 85–88], and in general has consequences for the propagation of information about a stimulus to subsequent processing stages [89]. Hence, it is important that future studies investigate how different network topologies can change both the signal, i.e. the effects of the single-cell stimulation, and the noise, i.e. the fluctuations in the network's activity.

## Methods

### Detailed description of the recurrent network model

**Single-neuron properties and total input to neurons.** We modeled all neurons as leaky integrate-and-fire point neurons [90]. The $k$th neuron follows the differential equation

$$\tau_{m,k}\frac{\mathrm{d}v_k}{\mathrm{d}t}(t) = -v_k(t) + R_{m,k}I_{\mathrm{total},k}(t), \tag{24}$$

where the membrane time constant $\tau_{m,k}$ was drawn from a lognormal distribution with mean $\tau_{m,e} = \tau_{m,s} = 20$ ms if $k$ is a RS neuron or a SOM-LTS neuron, or with mean $\tau_{m,i} = 10$ ms if $k$ is a FS neuron. The standard deviation of all three distributions was set to 20% of the mean. These values are compatible with experimental measurements for the rat barrel cortex [29, 31]. The membrane resistance is $R_{m,k} = \tau_{m,k}/C_m$, where the capacitance $C_m = 150$ pF is assumed equal for all neurons. Eq (24) is complemented with the rule that whenever $v_k(t)$ reaches the threshold value $v_{T,k}$, the neuron emits a spike and $v_k(t)$ is reset and clamped at $v_R = 10$ mV for the duration of the refractory period $\tau_{\mathrm{ref},k}$. The value of the firing threshold was drawn independently for each neuron from a Gaussian distribution [31] with mean $v_{T,E} = v_{T,I} = 20$ mV if $k$ is an RS or FS neuron [29, 31] and with mean $v_{T,S} = 14$ mV if the $k$th neuron belongs to the SOM-LTS population, because the distance from resting potential to threshold is 5 mV to 7 mV lower in SOM-LTS neurons than in RS and FS neurons [29]. The standard deviation was set to 10% of the mean for all three neuron types [29]. The refractory time is $\tau_{\mathrm{ref},k} = \tau_{\mathrm{ref},0} + \hat{\tau}_{\mathrm{ref},k}$, where $\tau_{\mathrm{ref},0} = 4$ ms and $\hat{\tau}_{\mathrm{ref},k}$ was drawn from a lognormal distribution with mean 2 ms and standard deviation 1 ms. The variability in the refractory time serves the purpose of mimicking the variability in the maximum firing rate of neurons [29].

If the $k$th neuron belongs to the FS population, its total input current $I_{\mathrm{total},k}$ is just the sum of the external input and of the recurrent input, i.e. it reads:

$$R_{m,k}I_{\mathrm{total},k}(t)\Big|_{k\in\mathrm{FS}} = R_{m,k}[I_{\mathrm{ext},k}(t) + I_{\mathrm{rec},k}(t)], \tag{25}$$

where the first term on the right side of Eq (25) represents the external input from other brain

areas and the second term models the recurrent local input from other neurons within the network. If the considered $k$th neuron belongs either to the RS or to the SOM-LTS population, the total input current includes an additional adaptation term $a_k(t)$:

$$R_{m,k}I_{\text{total},k}(t) = R_{m,k}[I_{\text{ext},k}(t) + I_{\text{rec},k}(t) - a_k(t)]. \qquad (26)$$
$$\scriptstyle k \in \text{RS, SOM}$$

The adaptation current in the last equation obeys [37, 38]:

$$\tau_{a,k}\frac{\mathrm{d}a_k}{\mathrm{d}t}(t) = -a_k(t) + \tau_{a,k}\Delta a_k x_k(t), \qquad (27)$$

where $x_k(t) = \sum_j \delta(t - t_{k,j})$ is the spike train emitted by neuron $k$. In other words, every time the neuron fires, the adaptation current jumps by $\Delta a_k$. Otherwise, it decays to zero with the time constant $\tau_{a,k}$.

Both $\Delta a_k$ and $\tau_{a,k}$ are randomly drawn from a lognormal distribution with standard deviation equal to 20% of the mean. For RS neurons, the mean of the two distributions are $\tau_{a,e} = 100$ ms and $\Delta a_e = 0.3$ nA, respectively; for SOM-LTS neurons they are $\tau_{a,s} = 50$ ms and $\Delta a_s = 0.2$ nA, respectively.

With this choice of parameters, the strength of the spike-frequency adaptation roughly agrees with *in vitro* measurements from the layer IV of the rat barrel cortex [29, 36].

**External input to the network.** The external input encompasses one constant term and two excitatory Poissonian shot-noise processes:

$$\begin{aligned}
R_{m,k}I_{\text{ext},k}(t) \quad &= R_{m,k}I_0 + \tau_{m,k}\left[\sum_{j=1}^{C_{\text{ext,th},k}}\sum_l J_{k,j,l}^{th}\delta(t - t_{k,j,l}^{th})\right. \\
&\left. + \sum_{p=1}^{C_{\text{ext,bc},k}}\sum_q J_{k,p,q}^{bc}\delta(t - t_{k,p,q}^{bc})\right].
\end{aligned} \qquad (28)$$

The constant term is set to $R_{m,k}I_0 = 10$ mV for all neurons. The second term in Eq (28) represents the input from the thalamus, and the third mimics incoming spikes from surrounding cortical areas. Because the thalamus has a higher average firing rate, "thalamic" input spikes at times $t_{k,j,l}^{th}$ occur at an average rate of $r_{\text{ext,th}} = 10$ Hz, while "cortical" input spikes at times $t_{k,p,q}^{bc}$ have a lower rate of $r_{\text{ext,bc}} = 2$ Hz. The number of external input spike trains depends on the cell type. Experimental studies suggest that SOM cells, in contrast to RS and FS cells, receive only weak input from the thalamus and from distant brain areas [29, 40]. Therefore, if the $k$th neuron belongs to the SOM-LTS population, then the number of external inputs is set to zero $C_{\text{ext,th},k} = C_{\text{ext,bc},k} = 0$, whereas when $k$ is a RS or a FS neuron, then $C_{\text{ext,th},k} = 500$. Furthermore, dendrites of FS neurons tend to be more localized than those of pyramidal cells, i.e. to receive more input from local RS neurons and less from distant ones. Hence, the number of inputs mimicking the cortical surroundings is $C_{\text{ext,bc},e} = 2000$ when $k$ is a RS neuron, and $C_{\text{ext,bc},i} = 1000$ when $k$ is a FS neuron. Each input spike causes a PSP drawn independently from an exponential distribution with mean $J_{\text{ext,e}} = 0.1$ mV when $k$ is a RS neuron, and from an exponential distribution with mean $J_{\text{ext,i}} = 0.2$ mV when $k$ is a FS neuron, because both thalamic and cortical excitatory postsynaptic potential (EPSP) amplitudes are larger in FS cells than in RS cells [29].

**Recurrent input to RS neurons.**   The recurrent input term $I_{\text{rec},k}(t)$ depends on the neuron type. If $k$ is a RS cell, it is

$$
\underset{k \in \text{RS}}{R_{m,k} I_{\text{rec},k}(t)} = \tau_{m,k} \left[ \sum_{i \in \mathcal{P}_e(k)} J_{ki}(t) x_i(t - D_{ki}) \right.
$$

$$
- \sum_{j \in \mathcal{P}_i(k)} J_{kj}(t) x_j(t - D_{kj}) \tag{29}
$$

$$
\left. - \sum_{\ell \in \mathcal{P}_s(k)} J_{k\ell}(t) x_\ell(t - D_{k\ell}) \right],
$$

where $x_i(t - D_{ki})$ indicates the spike train emitted by neuron $i$, $D_{ki}$ represents the total transmission delay resulting from the axonal propagation, the neurotransmitter diffusion, and the dendritic propagation from neuron $i$ to neuron $k$, $J_{ki}$ stands for the synaptic strength from neuron $i$ to neuron $k$, which depends on the spiking history (see below).

Connections to neuron $k$ originate from three sets of neurons: $\mathcal{P}_e(k)$, formed by $C_{ee} = 300$ randomly selected RS neurons, $\mathcal{P}_i(k)$, consisting of $C_{ei} = 200$ randomly selected FS neurons, and $\mathcal{P}_s(k)$, composed of $C_{es} = 100$ randomly selected SOM-LTS neurons. Hence, the connection probability of RS-to-RS synapses is $C_{ee}/N_e = 15\%$, of FS-to-RS and of SOM-LTS-to-RS is $C_{ei}/N_i = C_{es}/N_s = 50\%$, consistent with the experimental observations that the connections between RS cells are sparse whereas those between RS and inhibitory cells are dense [27, 29, 41, 91–93]. Transmission delays are drawn uniformly in the range 0.5 ms to 1.0 ms [93]. All synaptic weights in Eq (29) undergo short-term depression (STD)

$$
J_{ki}(t) = J_{ki} R_{ki}(t^-), \tag{30}
$$

where the maximum coupling amplitudes $J_{ki}$ (corresponding to the first spike after neuron $i$ has not been firing for a long time) are drawn independently from an exponential distribution with mean $J_{ee} = 0.1$ mV for RS-to-RS connections, $J_{ei} = 0.5$ mV for FS-to-RS coupling, and $J_{es} = 0.25$ mV for SOM-to-RS connections. The variables $R_{ki}(t)$ represent the fraction of available synaptic resources, and $t^-$ indicates that the function is evaluated immediately before a spike. Model and parameters of STD, i.e. the time evolution of the $R_{ki}(t)$, are described below.

**Recurrent input to FS neurons.**   The recurrent input to a FS neuron reads:

$$
\underset{k \in \text{FS}}{R_{m,k} I_{\text{rec},k}(t)} = \tau_{m,k} \left[ \sum_{i \in \mathcal{Q}_e(k)} J_{ki}(t) x_i(t - D_{ki}) \right.
$$

$$
- \sum_{j \in \mathcal{Q}_i(k)} J_{kj}(t) x_j(t - D_{kj})
$$

$$
- \sum_{p \in \mathcal{Q}_s(k)} J_{kp}(t) x_p(t - D_{kp}) \tag{31}
$$

$$
\left. + \sum_{\ell \in \text{FS}} \hat{J}_{k\ell} x_\ell(t - D_{k\ell}) \right],
$$

where the first term represents the synaptic input from $\mathcal{Q}_e(k)$, a set of $C_{ie} = 800$ randomly selected RS cells (connection probability $C_{ie}/N_e = 40\%$), the second term is the input from the inhibitory FS presynaptic population $\mathcal{Q}_i(k)$ with size $C_{ii} = 200$ (connection probability $C_{ii}/N_i = 50\%$), and the third term represents the inputs from $\mathcal{Q}_s(k)$, $C_{is} = 50$ randomly selected

SOM-LTS neurons (connection probability 25%). All weights appearing in these three terms follow Eq (30) and their peak value is drawn from an exponential distribution of mean $J_{ie} = 0.2$ mV, $J_{ii} = 1.0$ mV, and $J_{is} = 0.1$ mV, respectively. Transmission delays are the same as for RS-to-RS connections. These values reflect the fact that FS neurons receive strong and dense connections both from RS and from FS neurons, and that synapses from SOM to FS neurons are comparatively weaker [29, 33]. The fourth and last term in Eq (31) is an effective model for the electrical coupling among FS cells (gap junctions), see next subsection.

**Effective model for gap junctions.** Both FS and SOM neurons in the rat somatosensory cortex are coupled by gap junctions [32, 34, 40, 94]. In a simplified picture, gap junctions act as a passive conductance coupling between the membrane voltage of two neurons. The standard way of mimicking the effect of a gap junction between neuron $k$ and neuron $\ell$ would be the following additional current for neuron $k$ [95, 96]:

$$R_{m,k}I_{GJ,k\ell} = \gamma_{k\ell}(v_\ell - v_k) + \tau_{m,k}\hat{J}_{k\ell}x_\ell(t - D_{k\ell}), \tag{32}$$

where $\gamma_{k\ell}$ is proportional to the Ohmic conductance between the two neurons and modulates the strength of the subthreshold coupling, and $\hat{J}_{k\ell}$ models the effect of spikes fired by neuron $\ell$, which has to be added *ad hoc*, because LIF neurons do not explicitly generate action potentials. Gap junctions typically form between dendrites of different neurons. Therefore, the effect of spikes must travel from the soma along the dendrite of the first neuron to the gap junction and then from it along the dendrite into the soma of the second neuron. For this reason, the time necessary for this propagation can be as large as 0.5 ms [97]. Hence the delay term $D_{k\ell}$ is drawn from a uniform distribution in the range 0.1 ms to 0.5 ms. As reported in the main text, the subthreshold coupling is completely neglected here, i.e. $\gamma_{j\ell} = 0$ is set for all neuron pairs. The subthreshold coupling was shown to have a very weak influence on the firing rate, synchrony, and oscillation frequency of networks of LIF neurons, as opposed to the spike-related coupling [35]. The amplitude of gap-junction-related post-synaptic potentials measured in FS neurons of the rat somatosensory cortex is rather variable and, on average, about half as large as excitatory post-synaptic potentials induced by RS neurons [97, 98]. Hence, $\hat{J}_{k\ell}$ was drawn from an exponential distribution of mean $\hat{J}_{ii} = J_{ie}/2 = 0.05$ mV. The probability of a gap junction connecting two neighboring inhibitory neurons of the same type (FS with FS and SOM with SOM) is high (60% to 80% [40, 94]). For simplicity, the gap-junction coupling was approximated here as all-to-all (without self-coupling).

**Recurrent input to SOM-LTS neurons.** Finally, the recurrent input to a SOM-LTS neuron is

$$\begin{aligned}
R_{m,k}I_{\mathrm{rec},k}(t) \atop {\scriptstyle k\in\mathrm{SOM}} &= \tau_{m,k}\left[\sum_{i\in\mathcal{L}_e(k)} J_{ki}(t)x_i(t - D_{ki})\right.\\
&\quad - \sum_{j\in\mathcal{L}_i(k)} J_{kj}(t)x_j(t - D_{kj}) \\
&\quad \left.+ \sum_{\ell\in\mathrm{SOM}} \hat{J}_{k\ell}x_\ell(t - D_{k\ell})\right].
\end{aligned} \tag{33}$$

The three terms in Eq (33) represent the input from excitatory RS neurons, from inhibitory FS neurons, and from gap junctions, respectively. Gap-junctions are modeled in the same way as for FS neurons: their amplitudes and delays are drawn from the same distributions. The first term in Eq (33) is the input from $C_{se} = 1000$ randomly chosen RS neurons (connection

probability $C_{se}/N_e$ = 50%). These are the only connections that undergo short-term *facilitation* instead of depression, and for which random *transmission failures* were modeled (details on the model below). The static baseline amplitudes $J_{ki}$ of each synapse are drawn independently from an exponential distribution and have mean $J_{se}$ = 0.1 mV. The second term in Eq (33) represents the input from $C_{si}$ = 100 randomly selected FS neurons (connection probability $C_{si}/N_i$ = 25%). These connections have the average maximum strength $J_{si}$ = 0.25 mV, undergo short-term depression and obey Eq (30). Chemical synapses between SOM neurons are infrequent and weak [29, 40] and were omitted for simplicity.

**Model of short-term depression.** Except for those connecting RS to SOM neurons, all chemical synapses in the model undergo short-term depression. Each weight $J_{kj}(t)$ has a time dependence described by

$$J_{ki}(t) = J_{ki}R_{ki}(t^-), \tag{34}$$

where the variable $R_{ki}(t)$ stands for the fraction of available synaptic resources and is described by the standard model by Tsodyks and Markram [44],

$$\dot{R}_{ki}(t) = \frac{1 - R_{ki}(t)}{\tau_D} - U_{se}R_{ki}(t^-)\sum_j \delta(t - \hat{t}_{i,j}), \tag{35}$$

where $\hat{t}_{i,j}$ are the times at which the spikes of neuron $i$ arrive at the synapse, and $t^-$ indicates that the function is evaluated at $t - \varepsilon$ ($\varepsilon > 0$ is a small positive number), i.e. just before a spike. The parameter $U_{se}$ represents the release probability and $\tau_D$ is the recovery time scale. Note that the time evolution of $R_{ki}(t)$ depends on the spike times of the presynaptic neuron $i$ only. Hence, if $\tau_D$ and $U_{se}$ do not depend on $k$, the time course of each variable $R_{ki}(t)$ is a time-shifted copy of a single master variable $R_i(t)$

$$R_{ki}(t) = R_i(t - D_{ki}), \tag{36}$$

where $R_i(t)$ obeys the same equation as $R_{ki}(t)$, except that the arrival times $\hat{t}_{i,j}$ in Eq (35) are replaced by $t_{i,j}$, the spike times of neuron $i$. Here, it is assumed that $\tau_D$ and $U_{se}$ only depend on the type of the source and target neuron, but not on the identity of the particular neuron within a population so that Eq (36) holds. In this way, the actual number of dynamic variables required to simulate the network is reduced from one variable per synapse to one variable per neuron, which is an enormous computational advantage.

The parameter values chosen to model strong depression (all connections depicted in blue in Fig 1) are $\tau_D = \tau_{D,s}$ = 150 ms and $U_{se} = U_{se,s}$ = 0.2. With this choice, the eighth PSP of a 40 pre-synaptic regular spike train is about one half of the maximal amplitude [29]. Most chemical synapses in the barrel cortex are depressing ([29, 43, 69]). However, inhibitory synapses originating from SOM-LTS neurons and terminating onto RS neurons show only weak depression or slight facilitation. Here, these connections are modeled as mildly depressing (Fig 1, light blue) by setting $\tau_D = \tau_{D,w}$ = 50 ms and $U_{se,w}$ = 0.05. For simplicity, also SOM-to-FS connections were given the same STD parameters.

**Short-term facilitation and transmission failures.** Excitatory synapses from RS neurons to SOM-LTS neurons (depicted in red in Fig 1) are strongly facilitating ([29, 41, 42]). If the parameter $U_{se}$ in Eq (36) is turned into a dynamical variable, $u(t)$, facilitating synapses can be described [45, 99]. The amplitude of the PSPs is proportional to the product $R(t)u(t)$. Considering the connection from the RS neuron $i$ to the SOM-LTS neuron $k$, the time evolution of the synaptic amplitude is described by (note that the conventions have been slightly changed

with respect to ref. [45]):

$$J_{ki}(t) = J_{ki}R_i(t^- - D_{ki})\frac{u_i(t^+ - D_{ki})}{U_b},$$  (37)

where $t^+$ means that the function is evaluated at $t + \varepsilon$, i.e. the value of $u_i(t)$ just after the occurrence of a spike. The variables $R_i(t)$ and $u_i(t)$ obey

$$\dot{u}_i(t) = \frac{U_b - u_i(t)}{\tau_F} + (1 - u_i(t^-))U\sum_j \delta(t - t_{i,j})$$  (38)

$$\dot{R}_i(t) = \frac{1 - R_i(t)}{\tau_D} - u_i(t^-)R_i(t^-)\sum_j \delta(t - t_{i,j}),$$  (39)

where $t_{i,j}$ indicate the spike times of neuron $i$. The first term in Eq (38) governs the relaxation of the facilitation variable to the baseline level $U_b$ and the second term determines a positive jump upon each pre-synaptic spike. The time evolution of the depression variable $R_i(t)$ has the same form of Eq (35), i.e. a purely depressing synapse, except that the release probability is the time-dependent function $u_i(t)$. The choice of the parameters $U$, $\tau_F$, $\tau_D$ dictates whether, for a given firing rate, the synapse facilitates, depresses, or both [45]. Here, the four parameters appearing in Eqs (38) and (39) were set as follows: $\tau_F = 300$ ms, $\tau_D = \tau_{D,f} = 100$ ms, $U_b = 0.01$, and $U = 0.03$. With this choice and for a pre-synaptic stimulation of 40 Hz, the synapse is purely facilitating [29].

RS-to-SOM synapses stand out from all other synapses considered here because of a much higher occurrence of transmission failures at low presynaptic firing rates (the average failure rate is $\approx$10% for RS-to-RS synapses, $\approx$ 5% for synapses to and from FS neurons, and $\gtrsim$ 50 for RS to SOM-LTS synapses [29]). However, the failure rate of RS-to-SOM-LTS synapses decreases to $\approx$ 10% upon repeated stimulation at 40 (failure rates for other synapses weakly depend on the presynaptic firing rate [29]). Here, transmission failures are modeled only for RS-to-SOM synapses via a stochastic binary variable $S(p_f)$:

$$S(p_f) = \begin{cases} 1 & \text{with probability } 1 - p_f \\ 0 & \text{with probability } p_f \end{cases},$$  (40)

where $p_f$ describes the failure rate, which obeys the following dynamical equation

$$\dot{p}_f(t) = \frac{p_{f,\text{rest}} - p_f(t)}{\tau_f} - G(p_f, \Delta p_f, p_{\min})\sum_j \delta(t - t_{i,j}).$$  (41)

In the last equation, $p_{f,\text{rest}} = 0.5$ is the baseline failure rate. Upon each presynaptic spike, the failure rate decreases by $G(p_f, \Delta p_f, p_{\min})$ and relaxes back to the baseline value with the time constant $\tau_f = 250$ ms. The size of each downward jump is $\Delta p_f = 0.1$ but is constrained to values above $\Delta p_f = p_{\min} = 0.1$, a condition which is imposed by the piecewise linear function

$$G(p_f, \Delta p_f, p_{\min}) = \begin{cases} 0 & \text{if } p_f \leq p_{\min} \\ p_f - p_{\min} & \text{if } p_{\min} < p_f < p_{\min} + \Delta p_f . \\ \Delta p_f & \text{if } p_{\min} + \Delta p_f \leq p_f \end{cases}$$  (42)

In the end, the synaptic weight from the RS neuron $i$ to the SOM-LTS neuron $k$ obeys the following equation:

$$J_{ki}(t) = J_{ki}R_i(t^-  - D_{ki})\frac{u_i(t^+ - D_{ki})}{U_b}S\Big(p_{f,i}(t^- - D_{ki})\Big). \tag{43}$$

If the average effect of synaptic failures is taken into account, a 40 Hz presynaptic stimulation causes the eighth PSP to be about eight times larger than the first, which is in a reasonable qualitative agreement with the strong amplification measured *in vitro* [29, 42].

## Detailed description of the readout network model

The differentiator network readout (DNR) consists of one population of $N_B = 10000$ RS neurons ($\mathcal{S}^B$) and of one population of 2000 FS neurons ($\mathcal{I}$). Each neuron in the DNR follows the same dynamical equation as its counterpart within the BCN and receives feedforward input from $C^{\mathrm{read},E} = 1000$ randomly selected RS neurons of the BCN, from $C^{\mathrm{read},I} = 100$ randomly selected FS neurons of the BCN, and from $C^{\mathrm{read},S} = 100$ randomly selected SOM neurons of the BCN. In other words, the size of the presynaptic population of each excitatory and inhibitory neuron within the readout network is equal to the readout sets of the other detection schemes. The cellular properties of RS and FS neurons within the readout network are statistically equivalent to their counterparts within the BCN, except that the average strength and recovery time of the spike-frequency adaptation variable of RS neurons within the DNR was reduced to $\Delta a_e = 0.1$ nA and $\tau_{a,e} = 50$ ms, respectively (the relative standard deviation of these parameters was again 20%). A further difference is that the average rate of the Poisson input mimicking cortical input was reduced to 50% of that of the BCN (the "thalamic" random input is the same), and that each neuron in the DNR receives 200 random connections from the local inhibitory population ($\mathcal{I}$). Hence, the only recurrent connections within the DNR are inhibitory, as depicted in Fig 2C.

All connections from the BCN to the DNR and within the DNR are randomly drawn from the same distributions as for the corresponding class of neurons within the BCN, except for the connections from the inhibitory readout population $\mathcal{I}$ to the excitatory readout population $\mathcal{S}^B$, the average strength of which, $J_{ei}^{\mathrm{R}}$, is tuned to a value that enables the DNR to approximate the function of a differentiator circuit, as explained in the following.

Referring to Fig 4, we have to calculate the value of $J_{ei}^{\mathrm{R}}$ such that the input $\Delta\mu_{\mathcal{I}}$ via the indirect path to the readout population $\mathcal{S}^B$ equals a negative and temporally delayed image of the direct input $\Delta\mu_e$. This value can approximately be determined by the following linear-response calculation.

Consider a perturbation of the firing rate of the RS neurons within the BCN and indicate it with $\Delta r_e$. We assume that the perturbation is slow compared to the most important system time constants so that time-dependencies can be neglected. As a consequence of the firing rate perturbation within the BCN, the mean input from the BCN to $\mathcal{S}^B$ changes by

$$\Delta\mu_e = \tau_{m,e}J_{ee}^{\mathrm{FF}}\bar{R}(r_e)\hat{C}\Delta r_e, \tag{44}$$

where the term

$$\bar{R}(r) = \frac{1}{1 + \tau_{D,s}U_{se,s}r} \tag{45}$$

represents the average effect of the short-term depression (STD), given a presynaptic firing rate $r$. In Eq (44), $J_{ee}^{\mathrm{FF}}$ represents the average synaptic strength of the connections from the BCN

to the excitatory readout population $\mathcal{S}^B$. Likewise, the mean input from $\mathcal{I}$ to $\mathcal{S}^B$ changes by

$$\Delta\mu_{\mathcal{I}} = -\tau_{m,e} J_{ei}^{R} \bar{R}(r_{\mathcal{I}}) C_{ei}^{R} \Delta r_{\mathcal{I}}, \tag{46}$$

where $C_{ei}^{R} = 200$ is the number of input connections from $\mathcal{I}$ to $\mathcal{S}^B$ per postsynaptic neuron, and $\Delta r_{\mathcal{I}}$ is the change in the firing rate of the population $\mathcal{I}$ from the spontaneous value $r_{\mathcal{I}}$.

The linear-response approximation of $\Delta r_{\mathcal{I}}$ is

$$\Delta r_{\mathcal{I}} = \frac{\mathrm{d}\phi_{sn}}{\mathrm{d}\mu} \tau_{m,i} \left( J_{ie}^{FF} \bar{R}(r_e) \hat{C} \Delta r_e - J_{ii}^{R} \bar{R}(r_{\mathcal{I}}) C_{ii}^{R} \Delta r_{\mathcal{I}} \right), \tag{47}$$

where $\mathrm{d}\phi_{sn}/\mathrm{d}\mu$ is the so-called DC susceptibility, i.e. the linear response of the firing rate of a LIF neuron to a slow change in its total mean input $\mu$. The value of the DC susceptibility can be approximated by taking the derivative of the firing rate of a white-shot-noise-driven LIF neuron [100] with respect to its mean input. The explicit expression for $\mathrm{d}\phi_{sn}/\mathrm{d}\mu$ with a non-zero refractory period can be found in the first appendix of [50].

First, Eq (47) can be solved for $\Delta r_{\mathcal{I}}$ and substituted into Eq (46). Then, we require that the perturbation in the mean input from direct and indirect pathways cancel each other (see Fig 4). In other words, we impose $\Delta\mu_e + \Delta\mu_{\mathcal{I}} = 0$ and finally solve for $J_{ei}^R$, which yields

$$J_{ei}^{R} = \frac{J_{ee}^{FF} \left( 1 + \tau_{m,i} \frac{\mathrm{d}\phi_{sn}}{\mathrm{d}\mu} J_{ii}^{R} \bar{R}(r_{\mathcal{I}}) C_{ii}^{R} \right)}{\tau_{m,i} \frac{\mathrm{d}\phi_{sn}}{\mathrm{d}\mu} J_{ie}^{FF} C_{ei}^{R} \bar{R}(r_{\mathcal{I}})}. \tag{48}$$

The only unknown quantity on the right hand side of Eq (48) is $r_{\mathcal{I}}$, the spontaneous firing rate of $\mathcal{I}$. This firing rate can be estimated from the numerical solution of the following self-consistency condition:

$$r_{\mathcal{I}} = \phi_{sn}(J_{ee}^{FF}, J_{ii}^{R}, r_{tot}^{in}, C_{ii}^{R} \cdot r_{\mathcal{I}}, I_{ext}), \tag{49}$$

where $r_{tot}^{in} = \hat{C} r_e + C_{ext,bc,e}^{R} r_{ext,bc} + C_{ext,th,e}^{R} r_{ext,th}$ is the total excitatory input rate to $\mathcal{I}$ and $\phi_{sn}(a_e, a_i, R_e, R_i, I_0)$ is the firing rate of a LIF neuron driven by white shot-noise with exponentially distributed weights [100]. The first two arguments, $a_e$, $a_i$ are the excitatory and inhibitory mean input weights, respectively. The third and fourth argument $R_e$, $R_i$ are the input rates of the excitatory and inhibitory input, respectively. The last argument $I_0$ is the constant input. The explicit expression with non-zero input current and non-zero refractory period is

$$\phi_{sn}(a_e, a_i, R_e, R_i, I_0) = \left( \tau_{\mathrm{ref}} \right.$$
$$\left. + \tau_m \int_0^{1/a_e} \frac{\mathrm{d}s}{s} Z_0^{-1}(s) \left[ \frac{e^{s\hat{v}_T}}{1 - a_e s} - e^{s\hat{v}_R} \right] \right)^{-1} \tag{50}$$

where $\hat{v}_R = v_R - R_m I_0$, $\hat{v}_T = v_T - R_m I_0$, and $Z_0^{-1}(s) = (1 - a_e s)^{\tau_m R_e} (1 + a_i s)^{\tau_m R_i}$.

Substituting numerical values in Eq (48) reveals that $J_{ei}^R \approx 0.65\mathrm{mV}$ would satisfy the imposed condition. By choosing the smaller value $J_{ei}^R = 0.6\,\mathrm{mV}$, we obtain an imperfect compensation of the mean input, which, as shown above, ultimately leads to a good agreement with the experimental data.

### Experimental data

The experimental data appearing in Figs 7–9 are a part of the dataset of references [6, 48]. In particular, the data shown in Fig 7 are the average effect size (for each stimulus duration) of most cells shown in Fig 18A of reference [48] (800 ms stimuli were not used in the present study); the experimental effect size in Fig 8 is the average for each stimulus intensity (and duration) of the cells appearing in Fig 14A of reference [48]; the average effect size for regular and irregular stimulation of Fig 9 is based on the same dataset used for Fig 21C of reference [48]. For experimental procedures, we refer to [6, 48].

### Author Contributions

**Conceptualization:** Davide Bernardi, Guy Doron, Michael Brecht, Benjamin Lindner.

**Data curation:** Davide Bernardi, Guy Doron.

**Formal analysis:** Davide Bernardi, Benjamin Lindner.

**Funding acquisition:** Davide Bernardi, Michael Brecht, Benjamin Lindner.

**Investigation:** Davide Bernardi, Guy Doron.

**Project administration:** Benjamin Lindner.

**Resources:** Michael Brecht, Benjamin Lindner.

**Software:** Davide Bernardi.

**Supervision:** Michael Brecht, Benjamin Lindner.

**Validation:** Davide Bernardi.

**Visualization:** Davide Bernardi.

**Writing – original draft:** Davide Bernardi, Benjamin Lindner.

**Writing – review & editing:** Davide Bernardi, Guy Doron, Michael Brecht, Benjamin Lindner.

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
