## [Decision Letter · Decision Letter 0]

22 Jun 2020

Dear Dr. Bernardi,

Thank you very much for submitting your manuscript "A network model of the barrel cortex combined with a differentiator detector reproduces features of the behavioral response to single-neuron stimulation." for consideration at PLOS Computational Biology.

As with all papers reviewed by the journal, your manuscript was reviewed by members of the editorial board and by several independent reviewers. In light of the reviews (below this email), we would like to invite the resubmission of a significantly-revised version that takes into account the reviewers' comments.

Note that a common theme across reviews consists of questions the readout aspect of the model.  This aspect clearly needs some serious focus in the revision process.  In that process, please also be sure to clarify how this work represents a significant advance relative to the previous papers of this group. 

We cannot make any decision about publication until we have seen the revised manuscript and your response to the reviewers' comments. Your revised manuscript is also likely to be sent to reviewers for further evaluation.

Sincerely,

Jonathan Rubin

Associate Editor

PLOS Computational Biology

Wolfgang Einhäuser

Deputy Editor

PLOS Computational Biology

Reviewer's Responses to Questions

**Comments to the Authors:**

Reviewer #1: This paper examines the contribution of single-unit stimulation in a detailed computational model of barrel cortex.

MAJOR POINT (1)

A fundamental advantage of working with computational models is not just the ability to capture experimental results, but, going further, to test the robustness of these results to alterations in the design of the model itself. After all, in some respect this is something you can really only do with computational models, so you have to take advantage of it wherever possible. Unfortunately, authors present only limited advances in this respect. Here are three areas we would like to see extended:

(1) line 116: how large of a subset of RS cells are fed into the readout? how much does it affect readout performance (i.e., effect size)?

(2) how does performance deteriorate with delta_T<10 ms?

(3) line 314: would be nice to show a full curve of stim duration vs. number of spikes evoked w/ SEM. I expect this result should depend on the sparsity of connectivity, but should be largely independent of system size.

MAJOR POINT (2)

The model assumes that SOM and FS populations from the BCN network play no role in the readout of sensory information. While the effect of a single FS spike on FS population is small, SOM cells are strongly excited in response to stimulation. Therefore, could they contribute to readout accuracy? This could be easily addressed by feeding SOM to your DR or DNR readouts, and comparing the accuracy with readout from RS only.

MAJOR POINT (3)

When comparing the models (IR vs. DR vs. DNR) to experimental data, authors are strictly focusing on a measure of effect size. However, just as interesting, is the standard deviation (sigma_hat_x(t)) of the model in response to RS stimulation. This standard deviation is much larger for DNR than DR in all simulations from Figs.7,8,9. Which model fits the experimental data better? If this cannot be validated with available data, please discuss how it could be addressed. Further, what other comparisons could be make between DR and DNR with respect to to barrel cortex recordings?

MINOR POINTS

The legend of fig.2 is redundant, first line: A cell selected at random from the barrel cortex network (BCN) is selected at random. (Yes, we get it, it's selected at random...)

The way in which the detection problem is framed in Fig.3 is highly reminiscent of the Tempotron problem (Gutig & Sompolinsky 2006 Nat Neurosci). However, this work is not cited. Please cite this paper and briefly address any similarity with your work (in terms of the problem, and not the learning algorithm of Gutig, of course, since you don't have any long-term plasticity).

While Fig.3 begins by describing the detection problem as a boundary-crossing problem with a lower boundary set on the readout activity, we learn later that in fact both a lower and an upper boundary can be used (unless I am misunderstanding...) (i.e., line 178). There is a bit of confusion about whether any (or all) of your different readouts (IR, DR, DNR) can indeed make use of both an upper and lower boundary. Please clarify.

Figure 1, last line of caption: no need to repeat the values of mean firing rates since they are already listed in the main text (line 188).

Figure 5: legend should describe what are the shaded areas in RS, FS, and SOM populations. this is not entirely clear from text – does it refer to connectivity, or some measure of network activity? All I see is " spikes of the stimulated RS neuron reach a large fraction of the FS population" but I don't understand whether this refers strictly to a connection probability or to spiking activity, e.g., in RS causing depolarization in a number of FS spikes. If the latter is true, then not sure how you establish such causality.

line 417: typo: These stimuli, in accordance with the experimental procedure [6, 47], is a. *are a*

Figure 7 panel: First row is what? you state "standardized deviation from the spontaneous value of the time-dependent mean readout activity" but I thought this was just the mean, i.e., Eq.13. why "standardized deviation"? Same question for the second row, why is it "standardized deviation"?

The caption of Fig.10 needs more explanation; are these different cells or the same cell over several recordings?

Reviewer #2: The authors present a large scale LIF model of cortex that contains pyramidal cells as well as SOM and PV inhibitory cells. They tune the network to lie in a balanced state and then stimulate single pyramidal neurons in order to mimic recent experiments that are in the barrel cortex of mice. Since the experiments indicate that single neuron stimuli can affect the behavior, the authors created this large scale simulation and then try to design a way to detect the presence of the stimulus. Thus, they build a second network that takes one of three forms: integrator, differentiator (via delays) and differentiator network. There are merits in the paper, but also some flaws that need to be addressed. The present work is not terribly different from the authors previous efforts in 2017,2018 other than applying them to the barrel. The fit of the data is quite good for the DR and DNR but we don’t really know if the IR was optimally fit.

Major Points:

A number of papers related to barrel circuitry inhibition and differentiation are missing from the discussion here. The authors should discuss papers by Pinto, Simons and their coauthors as well as the old integrate and fire models of barrel cortex by Kyriazi and Simons. All these papers are concerned with the understanding of how inhibition is able to make the barrel responses into more like a differentiator than an integrator and thee authors wrote many papers on this several decades ago. Since the key feature in the present paper is a so-called differentiator circuit, and these old papers tuned their model to fit experimental data from layer 4 rat barrels, it seems that they need to be acknowledged.

Is their network a single barrel? Or is this supposed to be the full cortex. I suspect the former but in any case they should use more precise language. Is this layer 4 or supposed to be some other layer? The abbreviation BCN is never defined. It is unclear to me what makes this barrel cortex as opposed to any other cortex.

The whole concept of a readout circuit seems really contrived and weakens the proposed model substantially. In the old work of the Simons group, there was no need for such a circuit, it was all contained in the basic barrel network. I am having a lot of trouble figuring out what part of the barrel ctx each of these modules represents as they are not specified eg as layer IV or what. The primacy of this aspect of the model makes it foubtful to me that it will be a good model for cortex as it seems to be an artifice that is needed to make sense of the extremenely noisy outputs of the model.

Minor points

--- Page 9 ---

So is jread synaptic depression. I don’t understand how it is implemented in the context of the readout neurons

--- Page 12 ---

Again, I see nothing here that points specifically to barrel cortex. I recall some old papers by Latham that looked at the effects of deleting or adding a single spike to a balanced network which I think they include in the references. They should discuss the PRX 2012 paper by Monteforte and Wolf who describe similar effects.

the authors say:

"Furthermore, their average amplitude is smaller if compared to other connections and they are strongly depressing, so that the direct effect on the overall firing rate of the RS population is small."

How can you tell that the effect is small without excluding the strong SOM inhibition?

--- Page 20 ---

They talk about the readout network being trained. How would this be accomplished. Also, again, I really don't see where this readout network sits in the anatomy of the barrel cortex.

They say:

"A readout that integrates spiking in a sliding time window would experience difficulty in distinguishing the possibly small changes ... "

This seems easily testable in the model with very little work. I think the authors should show this is true rather than saving it for another paper.

--- Page 37 ---

Can you discuss why the threshold crossing in from below for the IR and from above in the DN networks and maybe show an example in fig 3 of what that looks like?

--- Page 43 ---

Is the mean stimulus in fig9 symmetric about t=200?

Reviewer #3: Summary

Bernadi and colleagues – create a quasi-realistic model of barrel cortex, to understand how stimulation of individual cells, in turn impact network interactions and how the networks output can be read out by distinct read out classes (integrator versus differentiator).

Major Concerns

Is the model focusing on what is happening in layer 4? It appears this way given the choice and proportion of neuronal phenotypes – be specific.

Why assume the readout is only getting inputs from excitatory cells, if the model is representing layer 4 – then anatomical data suggests that both excitatory and inhibitory neurons within the layer 4-barrel synapse onto neurons in the supragranular and infragranular layers.

The goal of the study is to describe the influence of a single neuron in most cases with a relatively short punctate stimulus, while a good starting point, it is not clear how useful this will be given the nature of inputs in natural settings which co-activate hundreds if not thousands of neurons within very short time windows.

Minor Issues

Line 64 first use of BCN define the abbreviation at this point it is not defined in the text until line 70

Lines 67-68 2600 neurons might exist within a barrel or a barrel column but many more are within the barrel cortex itself.

**Have all data underlying the figures and results presented in the manuscript been provided?**

Reviewer #1: No: n/a

Reviewer #2: Yes

Reviewer #3: Yes

PLOS authors have the option to publish the peer review history of their article (what does this mean?). If published, this will include your full peer review and any attached files.

Reviewer #1: Yes: Jean-Philippe Thivierge

Reviewer #2: No

Reviewer #3: Yes: Joshua C. Brumberg
---

## [Decision Letter · Decision Letter 1]

30 Dec 2020

Dear Dr. Bernardi,

Thank you very much for submitting your manuscript "A network model of the barrel cortex combined with a differentiator detector reproduces features of the behavioral response to single-neuron stimulation." for consideration at PLOS Computational Biology. As with all papers reviewed by the journal, your manuscript was reviewed by members of the editorial board and by several independent reviewers. The reviewers appreciated the attention to an important topic. Based on the reviews, we are likely to accept this manuscript for publication, providing that you modify the manuscript according to the review recommendations.

Sincerely,

Jonathan Rubin

Associate Editor

PLOS Computational Biology

Wolfgang Einhäuser

Deputy Editor

PLOS Computational Biology

[LINK]

Reviewer's Responses to Questions

**Comments to the Authors:**

Reviewer #1: I thank the authors for their point-by-point reply to my comments. You have done a good job addressing my concerns in your revised work. I have mostly minor comments at this point, as detailed below.

Minor comments:

Line 133: The third readout scheme (fig. 2C) *is* based on the summed activity

Line 149: just to be clear, you mean that all dynamic weights J^e_read,i(t), J^i_read,i(t), and J^s_read,i(t) are depressing? please clarify.

Fig.3 legend: it is unclear what the vertical dashed lines at 0 ms and 600 ms indicate. I assume this is your "detection time window" [0,T_w] as mentioned in the main text, but this should be clarified in the legend.

Fig.4: on the figure itself, should delta_r be changed to delta_r_e to reflect the fact that the DRN perturbs RS neurons from the BCN? Further, to facilitate reading, I would suggest that the "strength of the connection from I to S^B" mentioned in the figure legend be followed by "(J^R_ei)".

Fig.5 legend: " as it can be seen by excluding" remove "it" (same comment for line 388)

Line 442: The sentence beginning with "As the most prominent features..." seems to be incomplete. I believe that what you mean is that there is a null net effect on effect size?

Fig.9, Panel A: the y-axis label should be "Rel. current intensity"

Fig.12, legend: "filtered with with a 1 s sliding window" remove "with"

Line 582: "justacelluar" should be "juxtacelluar"

Line 688: " differentiating would be a strategy to the detector uses": replace "to" with "that"

Discussion: I was surprised not to see a reference to the work of Mainen and Sejnowski 1995 Science, as they were amongst the first to discuss cortical responses to constant vs. irregular stimulation and its relation to spike reliability. Perhaps I am just getting old, but it would be nice to see at least a quick mention of this paper (perhaps around line 640 regarding general mechanisms across cortical areas?)

Reviewer #2: I was not able to read this easily as the figures are not in the main body of the text and are not even with the captions. It seems that you have answered my objections for the most part

Reviewer #3: the authors have effectively answered my concerns

**Have all data underlying the figures and results presented in the manuscript been provided?**

Reviewer #1: Yes

Reviewer #2: Yes

Reviewer #3: Yes

PLOS authors have the option to publish the peer review history of their article (what does this mean?). If published, this will include your full peer review and any attached files.

Reviewer #1: **Yes: **Jean-Philippe Thivierge

Reviewer #2: No

Reviewer #3: **Yes: **Joshua C. Brumberg
---

## [Editor Report · Decision Letter 2]

17 Jan 2021

Dear Dr. Bernardi,

We are pleased to inform you that your manuscript 'A network model of the barrel cortex combined with a differentiator detector reproduces features of the behavioral response to single-neuron stimulation.' has been provisionally accepted for publication in PLOS Computational Biology.

Best regards,

Jonathan Rubin

Associate Editor

PLOS Computational Biology

Wolfgang Einhäuser

Deputy Editor

PLOS Computational Biology

---

## [Editor Report · Acceptance letter]

2 Feb 2021

PCOMPBIOL-D-20-00502R2 

A network model of the barrel cortex combined with a differentiator detector reproduces features of the behavioral response to single-neuron stimulation.

Dear Dr Bernardi,

I am pleased to inform you that your manuscript has been formally accepted for publication in PLOS Computational Biology. Your manuscript is now with our production department and you will be notified of the publication date in due course.

With kind regards,

Alice Ellingham
